# Diversity Matters When Learning From Ensembles

**Giung Nam**[1*]**, Jongmin Yoon**[1*]**, Yoonho Lee**[2,3]**, Juho Lee**[1,2]
KAIST[1], Daejeon, South Korea, AITRICS[2], Seoul, South Korea, Stanford University[3], USA
{giung, jm.yoon, juholee}@kaist.ac.kr

## Abstract

Deep ensembles excel in large-scale image classification tasks both in terms of prediction accuracy and calibration. Despite being simple to train, the computation and memory cost of deep ensembles limits their practicability. While some recent works propose to distill an ensemble model into a single model to reduce such costs, there is still a performance gap between the ensemble and distilled models. We propose a simple approach for reducing this gap, i.e., making the distilled performance close to the full ensemble. Our key assumption is that a distilled model should absorb as much function diversity inside the ensemble as possible. We first empirically show that the typical distillation procedure does not effectively transfer such diversity, especially for complex models that achieve near-zero training error. To fix this, we propose a perturbation strategy for distillation that reveals diversity by seeking inputs for which ensemble member outputs disagree. We empirically show that a model distilled with such perturbed samples indeed exhibits enhanced diversity, leading to improved performance.

## 1  Introduction

Deep Ensemble (DE) [Lakshminarayanan et al., 2017], a simple method to ensemble the same model trained multiple times with different random initializations, is considered to be a competitive method for various tasks involving deep neural networks both in terms of prediction accuracy and uncertainty estimation. Several works have tried to reveal the secret behind DE's effectiveness. As stated by Duvenaud et al. [2016] and Wilson and Izmailov [2020], DE can be considered as an approximate Bayesian Model Average (BMA) procedure. Fort et al. [2019] studied the loss landscape of DEs and showed that the effectiveness comes from the diverse modes reached by ensemble members, making it well suited for approximating BMA. It is quite frustrating that most sophisticated approximate Bayesian inference algorithms, especially the ones based on variational approximations, are not as effective as DEs in terms of exploring various modes in parameter spaces.

Despite being simple to train, DE incurs significant overhead in inference time and memory usage. It is therefore natural to develop a way to reduce such costs. An example of such a method is Knowledge Distillation (KD) [Hinton et al., 2015], which transfers knowledge from a large *teacher network* to a relatively smaller *student network*. The student network is trained with a loss that encourages it to copy the outputs of the teacher networks evaluated at the training data. With KD, there have been several works that learn a single student network by distilling from DE teacher networks. A naïve approach would be to directly distill ensembled outputs of DE teachers to the single student network. A better way proven to be more effective is to set up a student network having multiple subnetworks (multiple heads [Tran et al., 2020] or rank-one factors [Mariet et al., 2020]) and distill the outputs of each ensemble member to each subnetwork in a one-to-one fashion. Nevertheless, the empirical performance of such distilled networks is still far inferior to DE teachers.

---

[*] Equal contribution

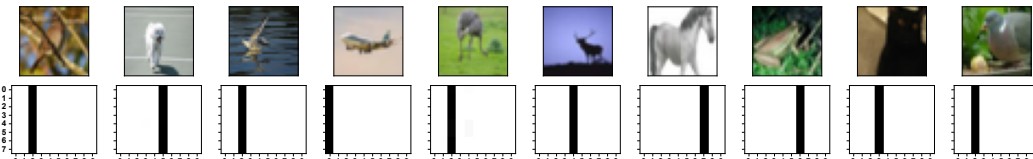

**Figure 1:** It shows randomly selected train examples from CIFAR-10 and their corresponding predictions given by 8 ResNet-32 models. In the second row, the vertical axis denotes the ensemble members, and the horizontal axis denotes the class index.

There may be many reasons why the students are not doing as well as DE teachers. We argue that a critical limitation of current distillation schemes is that they do not effectively transfer the diversity of DE teacher predictions to students. Consider an ensemble of deep neural networks trained for an image classification task. Given sufficient network capacity, each ensemble member is likely to achieve near-zero training error, meaning that the resulting outputs of ensemble members evaluated at the training set will be nearly identical, as shown in Fig. 1. In such a situation, reusing the training data during distillation will not encourage the student to produce diverse predictions. This is quite critical, considering various results establishing that the effectiveness of DE comes from averaging out its diverse predictions.

To this end, we propose a method that amplifies the diversity of students learning from DE teachers. Our idea is simple: instead of using the same training set, use a perturbed training set for which the predictions of ensemble members are likely to be diversified. To implement this idea, we employ Output Diversified Sampling (ODS) [Tashiro et al., 2020], a sampling scheme that finds small input perturbations that result in significant changes in outputs. Empirically, we confirm that the inputs perturbed with ODS result in large deviations in the DE teacher outputs. We further justify our method by analyzing the role of ODS perturbation. Specifically, we show that distilling using training data perturbed with ODS can be interpreted as an approximate Jacobian matching procedure where ODS improves the sample efficiency of the approximate Jacobian matching. Intuitively, by approximately matching Jacobians of teachers and students, we are transferring the outputs of teacher networks evaluated not only on the training data points but also on nearby points.

Using standard image classification benchmarks, we empirically validate that our distillation method promotes diversities in student network predictions, leading to improved performance, especially in terms of uncertainty estimation.

## 2 Backgrounds

### 2.1 Settings and notations

The focus of this paper is on the $K$-way classification problem taking $D$-dimensional inputs. We denote a student neural network as $\mathcal{S}(\boldsymbol{x}) : \mathbb{R}^D \to [0,1]^K$ and a teacher neural network as $\mathcal{T}(\boldsymbol{x}) : \mathbb{R}^D \to [0,1]^K$. $\mathcal{S}(\boldsymbol{x})$ and $\mathcal{T}(\boldsymbol{x})$ outputs class probabilities, and we denote the logits before softmax as $\hat{\mathcal{S}}(\boldsymbol{x})$ and $\hat{\mathcal{T}}(\boldsymbol{x})$. The $k$th element of an output is denoted as $\mathcal{S}^{(k)}(\boldsymbol{x})$. For DE teachers with $j = 1, \ldots, M$ members, the $j$th ensemble member is denoted as $\mathcal{T}_j(\boldsymbol{x})$. For a student network having $M$ subnetworks, the $j$th subnetwork is denoted as $\mathcal{S}_j(\boldsymbol{x})$.

### 2.2 Knowledge distillation

Knowledge distillation (KD) [Hinton et al., 2015] aims to train a student network $\mathcal{S}(\boldsymbol{x})$ by matching its outputs to the outputs of a teacher network $\mathcal{T}(\boldsymbol{x})$. In addition to the standard cross-entropy loss between student outputs and the ground-truth labels, *i.e.*, $\mathcal{L}_{\text{CE}}(\mathcal{S}(\boldsymbol{x}), \boldsymbol{y}) = -\sum_{k=1}^{K} y^{(k)} \log \mathcal{S}^{(k)}(\boldsymbol{x})$, KD additionally encourages the student's output to be as close as possible to that of the teacher by also minimizing the KL-divergence between output class distributions of the student and the teacher:

$$\mathcal{L}_{\text{KD}}(\mathcal{S}(\boldsymbol{x}), \mathcal{T}(\boldsymbol{x}); \tau) = -\tau^2 \sum_{k=1}^{K} \text{softmax}^{(k)}\left(\frac{\hat{\mathcal{T}}(\boldsymbol{x})}{\tau}\right) \log \text{softmax}^{(k)}\left(\frac{\hat{\mathcal{S}}(\boldsymbol{x})}{\tau}\right). \tag{1}$$

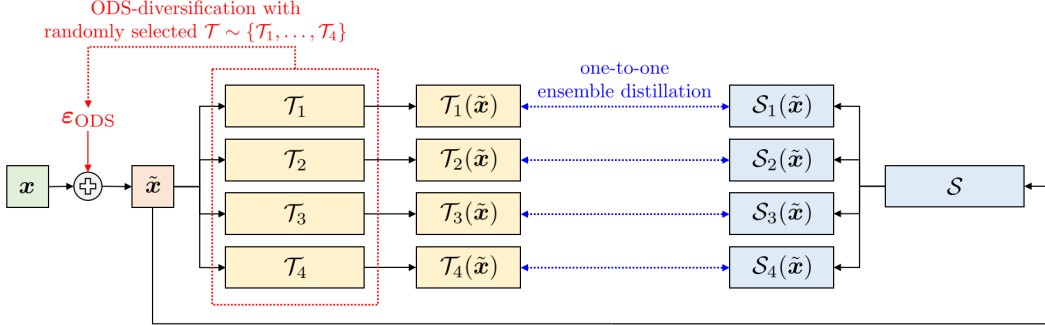

**Figure 2:** Overall structure of our method: (1) We first perturb train examples with respect to deep ensemble (DE) teachers, *i.e.*, $\boldsymbol{x} \to \tilde{\boldsymbol{x}}$. (2) DE teachers produce *diverse* predictions given perturbed train examples. (3) Since the diversity of teachers has been revealed, a student can effectively learn its diversity in the knowledge distillation framework.

Here, $\tau > 0$ is a *temperature* parameter which controls the smoothness of the distributions. We scale the KL divergence by $\tau^2$ since the soft targets scale as $1/\tau^2$. As a result, in the KD framework, the student imitates the teacher's outputs for training examples by minimizing $\mathcal{L} = (1 - \alpha)\mathcal{L}_{\text{CE}} + \alpha\mathcal{L}_{\text{KD}}$, where $\alpha \in (0, 1)$ is a hyperparameter to be specified.

## 2.3 BatchEnsemble and one-to-one distillation

BatchEnsemble (BE) [Wen et al., 2020] is a lightweight ensemble method that reduces the number of parameters by weight sharing. Specifically, each layer of a BE model consists of shared weights $\boldsymbol{W}$ and rank-one factors $\boldsymbol{r}_j \boldsymbol{s}_j^\top$ for $j = 1, \ldots, M$ with $M$ being number of subnetworks. Unlike the full ensemble that constructs full set of $M$ weights, BE shares $\boldsymbol{W}$ across all subnetworks while the rank-one factors are kept specific to each subnetwork. The weight matrix of the $j$th subnetwork is computed as $\boldsymbol{W} \circ \boldsymbol{r}_j \boldsymbol{s}_j^\top$ and consequently BE requires far fewer parameters than a full ensemble.

During training, BE simultaneously trains all subnetworks by feeding a duplicate of the current minibatch for each rank-one factor $\boldsymbol{r}_j \boldsymbol{s}_j^\top$ into the network. This procedure is easily vectorized because the computation graph for each subnetwork has the same structure, differing only in weight values. Mariet et al. [2020] proposed to further improve the performance of BE by distilling from fully trained DE teachers instead of training from scratch. During distillation, the parameters of $j$th subnetwork $\mathcal{S}_j(\boldsymbol{x})$ is trained by distilling from the $j$th DE teacher $\mathcal{T}_j(\boldsymbol{x})$. The corresponding objective function is then written as (c.f., Fig. 2 without perturbation)

$$\mathcal{L} = \sum_{j=1}^{M} \Big\{ (1 - \alpha)\mathcal{L}_{\text{CE}}(\mathcal{S}_j(\boldsymbol{x}), \boldsymbol{y}) + \alpha\mathcal{L}_{\text{KD}}(\mathcal{S}_j(\boldsymbol{x}), \mathcal{T}_j(\boldsymbol{x}); \tau) \Big\}. \tag{2}$$

Throughout the rest of the paper, we will use the BE as student networks and follow this one-to-one distillation framework.

## 2.4 Output diversified sampling (ODS)

ODS [Tashiro et al., 2020] is a sampling method that can maximize diversities in outputs of a given function; it is originally introduced to replace random input perturbation to enhance the performance of the adversarial attack. Let $\mathcal{F} : \mathbb{R}^D \to \mathbb{R}^K$ be a target function. ODS draws a perturbation vector guided from the *gradient* of the target function as follows.

$$\boldsymbol{w} \sim \text{Unif}([-1, 1])^K, \quad \varepsilon_{\text{ODS}}(\boldsymbol{x}, \mathcal{F}, \boldsymbol{w}) = \frac{\nabla_{\boldsymbol{x}}(\boldsymbol{w}^\top \mathcal{F}(\boldsymbol{x}))}{\|\nabla_{\boldsymbol{x}}(\boldsymbol{w}^\top \mathcal{F}(\boldsymbol{x}))\|_2} \in \mathbb{R}^D. \tag{3}$$

The intuition behind this sampling strategy is to seek the direction in the input space that maximizes the similarity between the function output and a randomly sampled vector $\boldsymbol{w}$. Following such a direction can make diverse output shifts due to the randomness in $\boldsymbol{w}$.

# 3 Learning from Ensembles with Output Diversification

## 3.1 One-to-one distillation with ODS

As discussed before, vanilla one-to-one distillation evaluates DE teachers on points where their outputs are nearly identical, so the students trained using them tend to exhibit low diversity in outputs. To avoid this undesirable behavior, we propose to instead train students on points perturbed with ODS. Let $(\boldsymbol{x}, \boldsymbol{y})$ be an training input pair. We first pick a *random teacher* $\mathcal{T}_r$ uniformly from $\{\mathcal{T}_1, \ldots, \mathcal{T}_M\}$ and perturb $\boldsymbol{x}$ as

$$\tilde{\boldsymbol{x}} = \boldsymbol{x} + \eta \varepsilon_{\mathrm{ODS}}(\boldsymbol{x}, \mathrm{softmax}(\hat{\mathcal{T}}_r(\boldsymbol{x})/\tau), \boldsymbol{w}), \tag{4}$$

where $\eta > 0$ is a step-size. Ideally, we would like to evaluate ODS vectors for each teacher to build $M$ perturbed versions of $\boldsymbol{x}$, but this would take too much time for gradient computation, especially when we have a large number of ensemble models. Instead, we just pick one of the teachers, use it to perturb the input, and share the perturbed input to train all students with the following loss function,

$$\mathcal{L} = \sum_{j=1}^{M} \Big\{ (1-\alpha)\mathcal{L}_{\mathrm{CE}}(\mathcal{S}_j(\boldsymbol{x}), \boldsymbol{y}) + \alpha \mathcal{L}_{\mathrm{KD}}(\mathcal{S}_j(\tilde{\boldsymbol{x}}), \mathcal{T}_j(\tilde{\boldsymbol{x}}); \tau) \Big\}, \tag{5}$$

where we are evaluating the KL-divergence between teacher outputs and student outputs on the *perturbed input*.

An implicit assumption here is that *a diversifying direction computed from a specific random teacher can bring diversities in the outputs of all the teacher networks*. This is based on the *transferability* assumption empirically justified in Tashiro et al. [2020]: ODS perturbations are computed on a surrogate model when we do not have access to the true model, assuming that the Jacobians computed from the surrogate model can approximate the true Jacobian to some extent. In our case, let $\mathcal{T}_1$ be a randomly picked teacher with which an ODS perturbation is computed, and $\mathcal{T}_2$ be another teacher. Assuming the transferability of Jacobian, we can let $\nabla_{\boldsymbol{x}} \mathcal{T}_1(\boldsymbol{x}) \approx \boldsymbol{R} \nabla_{\boldsymbol{x}} \mathcal{T}_2(\boldsymbol{x})$ for some matrix $\boldsymbol{R}$. Then the ODS perturbation computed from $\mathcal{T}_1(\boldsymbol{x})$ is

$$\varepsilon_{\mathrm{ODS}}(\boldsymbol{x}, \mathcal{T}_1, \boldsymbol{w}) \propto \boldsymbol{w}^\top \nabla_{\boldsymbol{x}} \mathcal{T}_1(\boldsymbol{x}) \approx (\boldsymbol{R}^\top \boldsymbol{w})^\top \nabla_{\boldsymbol{x}} \mathcal{T}_2(\boldsymbol{x}) \propto \varepsilon_{\mathrm{ODS}}(\boldsymbol{x}, \mathcal{T}_2, \boldsymbol{R}^\top \boldsymbol{w}). \tag{6}$$

That is, the ODS perturbation computed from $\mathcal{T}_1$ acts as another ODS perturbation with different random guide vector $\boldsymbol{R}^\top \boldsymbol{w}$. Hence, the same perturbation drives the outputs of $\mathcal{T}_1$ and $\mathcal{T}_2$ towards different directions. Note that without transferability, the Jacobians $\nabla_{\boldsymbol{x}} \mathcal{T}_1(\boldsymbol{x})$ and $\nabla_{\boldsymbol{x}} \mathcal{T}_2(\boldsymbol{x})$ would be vastly different and so are the directions $\boldsymbol{w}$ and $\boldsymbol{R}^\top \boldsymbol{w}$. This may be good in terms of diversification, but the resulting perturbed input act as an adversarial example not very useful for distillation, making the teachers completely disagree (refer to Appendix A.2 for more details). Our experiments in Section 5.2 and Section 5.3 empirically re-confirm this transferability assumption.

We can also interpret the random teacher selection as a stochastic approximation. Specifically, to find a direction maximizing diversities between teachers, we can consider an objective

$$\mathbb{E}_{\boldsymbol{w}_1, \ldots, \boldsymbol{w}_M} \left[ \sum_{j=1}^{M} \boldsymbol{w}_j^\top \mathcal{T}_j(\boldsymbol{x}) \right], \tag{7}$$

with $\boldsymbol{w}_1, \ldots, \boldsymbol{w}_M \overset{\text{i.i.d.}}{\sim} \mathrm{Unif}([-1,1])^K$. That is, we want to find a direction in the input space making the teachers $\{\mathcal{T}_j\}_{j=1}^{M}$ follow different guide vectors $\{\boldsymbol{w}_j\}_{j=1}^{M}$. Picking a random teacher and computing the ODS perturbation can be understood as a stochastic approximation of the gradient of this objective. Again, thanks to the transferability of the Jacobians, the variance of this stochastic gradient would be small, making it sufficient to use a single random teacher.

In addition, we empirically found that scaling the ODS updates according to confidence slightly improves performance. That is, we perturb an input $\boldsymbol{x}$ as

$$\tilde{\boldsymbol{x}} = \boldsymbol{x} + \eta C_{\max}(\boldsymbol{x}, \mathcal{T}_r, \tau) \varepsilon_{\mathrm{ODS}}(\boldsymbol{x}, \mathrm{softmax}(\hat{\mathcal{T}}_r(\boldsymbol{x})/\tau), \boldsymbol{w}), \tag{8}$$

where $C_{\max}(\boldsymbol{x}, \mathcal{T}_r, \tau) = \max_k \mathrm{softmax}^{(k)}(\hat{\mathcal{T}}_r(\boldsymbol{x})/\tau)$ denotes the maximum class probability. This scheme takes larger steps towards ODS directions for datapoints with high confidence. We call this modification ConfODS.

The training procedure with ODS perturbation is straightforward (Algorithm 1); we can just add the ODS computation procedure and replace the loss function from the original KD training procedure. Fig. 2 depicts our overall training procedure as a diagram.

---

**Algorithm 1** Knowledge distillation from deep ensembles with ODS perturbations

---

**Require:** Training data $\mathcal{D} = \{(\boldsymbol{x}_i, \boldsymbol{y}_i)\}_{i=1}^N$.
**Require:** Knowledge distillation weight $\alpha$, temperature $\tau$, learning rate $\beta$, ODS step-size $\eta$.
**Require:** Deep ensemble teacher networks $\{\mathcal{T}_j\}_{j=1}^M$ where $M$ denotes the size of ensembles.
   Let $\mathcal{S}$ be a student network which consists of $M$ subnetworks $\{\mathcal{S}_j\}_{j=1}^M$.
   Initialize parameters $\boldsymbol{\theta}$ of the student network $\mathcal{S}$.
   **while** not converged **do**
      Sample $(\boldsymbol{x}, \boldsymbol{y}) \sim \mathcal{D}$, $\boldsymbol{w} \sim \text{Unif}([-1, 1])^K$, and a teacher index $r$ uniformly from $\{1, \ldots, M\}$.
      Sample diversified inputs $\tilde{\boldsymbol{x}} = \boldsymbol{x} + \eta \boldsymbol{\varepsilon}_{\text{ODS}}(\boldsymbol{x}, \text{softmax}(\hat{\mathcal{T}}_r(\boldsymbol{x})/\tau), \boldsymbol{w})$
      $\mathcal{L}_{\text{CE}} \leftarrow -\sum_{j=1}^M \sum_{k=1}^K y^{(k)} \log \mathcal{S}_j^{(k)}(\boldsymbol{x})$
      $\mathcal{L}_{\text{KD}} \leftarrow -\tau^2 \sum_{j=1}^M \sum_{k=1}^K \text{softmax}^{(k)}(\hat{\mathcal{T}}(\tilde{\boldsymbol{x}})/\tau) \log \text{softmax}^{(k)}(\hat{\mathcal{S}}_j(\tilde{\boldsymbol{x}})/\tau)$
      $\boldsymbol{\theta} \leftarrow \boldsymbol{\theta} - \beta \nabla_{\boldsymbol{\theta}}((1-\alpha)\mathcal{L}_{\text{CE}} + \alpha\mathcal{L}_{\text{KD}})$
   **end while**

---

## 3.2 Interpretation as approximate Jacobian matching

We can also interpret our distillation strategy as an approximate Jacobian matching procedure. Srinivas and Fleuret [2018] shows that a KD procedure on inputs perturbed by small noise implicitly encourages matching the Jacobians of a teacher and a student. More specifically, let $\mathcal{S}$ and $\mathcal{T}$ be student and teacher models. The first-order Taylor expansion of the expected KD loss on perturbed inputs can be written as[1]

$$\mathbb{E}_{\boldsymbol{\varepsilon}}[\mathcal{L}_{\text{KD}}(\mathcal{S}(\boldsymbol{x}+\boldsymbol{\varepsilon}), \mathcal{T}(\boldsymbol{x}+\boldsymbol{\varepsilon})] = \mathcal{L}_{\text{KD}}(\mathcal{S}(\boldsymbol{x}), \mathcal{T}(\boldsymbol{x})) + \mathcal{L}_{\text{JM}}(\mathcal{S}(\boldsymbol{x}), \mathcal{T}(\boldsymbol{x})) + o(\sigma^2), \quad (9)$$

where $\sigma^2 = \mathbb{E}[\boldsymbol{\varepsilon}^\top \boldsymbol{\varepsilon}]$ and

$$\mathcal{L}_{\text{JM}}(\mathcal{S}(\boldsymbol{x}), \mathcal{T}(\boldsymbol{x})) = -\mathbb{E}_{\boldsymbol{\varepsilon}}\left[\sum_{k=1}^K \frac{\boldsymbol{\varepsilon}^\top \nabla_{\boldsymbol{x}} \mathcal{T}^{(k)}(\boldsymbol{x}) \nabla_{\boldsymbol{x}}^\top \mathcal{S}^{(k)}(\boldsymbol{x}) \boldsymbol{\varepsilon}}{\mathcal{S}^{(k)}(x)}\right]. \quad (10)$$

While Srinivas and Fleuret [2018] suggested using isotropic Gaussian noise $\boldsymbol{\varepsilon} \sim \mathcal{N}(\boldsymbol{0}, \sigma^2 \boldsymbol{I}_D)$, we argue that simple Gaussian noise is ineffective, especially for high-dimensional data that are likely to be embedded in low-dimensional manifolds. In such a case, most perturbation directions are orthogonal to the data manifold, and thus Gaussian noise becomes uninformative. On the other hand, ODS directly uses the Jacobian of the teacher to construct the perturbation direction. We conjecture that due to this fact, KD with ODS perturbation is a much more sample-efficient approximate Jacobian matching procedure compared to using Gaussian perturbations. We perform experiments to empirically validate this claim in Section 5.3.

The above analysis is for the usual KD framework with a single teacher and a single student. In our case, we are matching $M$ teachers to $M$ student subnetworks, but with ODS perturbations computed from a single teacher randomly sampled from $M$ teachers. Again, we are operating under the assumption that the Jacobian computed from a specific teacher will transfer well to other teachers, so the approximate Jacobian matching using it is still more effective compared to simply using Gaussian perturbations.

## 4 Related Works

**Knowledge distillation** KD is a method for transferring the information inside a large model into a smaller one [Hinton et al., 2015] by minimizing the KL divergence between teacher and student networks. [Wang et al., 2020] finds that additionally augmenting the data results in further performance gains because such augmentations allow the KD loss to tap into teacher information outside of the training set. Furthermore, modifications to the standard KD objective allow one to distill the knowledge in an ensemble of teachers into one student network while preserving the benefits in uncertainty estimation of ensembles [Malinin et al., 2020, Tran et al., 2020]. KD can be enhanced via gradient matching, and several existing methods [Srinivas and Fleuret, 2018, Czarnecki et al., 2017] propose sampling methods for knowledge distillation up to higher-order gradients. However, such existing methods suffer from a high computational burden or inefficient random perturbation.

---

[1]We assume $\tau = 1$ for notational simplicity, but the argument applies equally well to any $\tau$.

**Ensembles**  Ensemble methods [Hansen and Salamon, 1990, Dietterich, 2000], which construct a set of learners and make predictions on new data points by a weighted average, have been studied extensively. Ensembles perform at least as well as each individual member [Krogh and Vedelsby, 1994], and achieve the best performance when each member makes errors independently. Aside from benefits in model accuracy, DE [Lakshminarayanan et al., 2017, Ovadia et al., 2019] have recently shown to be a simple and scalable alternative to Bayesian neural networks because of their superior uncertainty estimation performance. Wen et al. [2020] proposes a network architecture that uses a rank-one parameterization to train an ensemble with low computation and memory cost.

## 5  Experiments

In this section, we try to answer the following questions with empirical validation.

- Do images perturbed by ODS increase diversity in DE teacher predictions? - Section 5.2.
- Do ODS perturbations encourage Jacobian matching? - Section 5.3.
- Does enhanced diversity of students lead to improved performance in terms of prediction accuracy and uncertainty estimation? - Section 5.4 and Appendix A.1.

### 5.1  Experimental setup

**Datasets and networks**  We compared our methods on CIFAR-10/100 and TinyImageNet. We used ResNet-32 [He et al., 2016] for CIFAR-10, and WideResNet-28x10 [Zagoruyko and Komodakis, 2016] for CIFAR-100 and TinyImageNet. Please refer to Appendix B.1 for detailed training settings.

**Hyperparameter settings**  The important hyperparameters for KD are the pair $(\alpha, \tau)$; for CIFAR-10, after a through hyperparameter sweep, we decided to stay consistent with the convention of $(\alpha, \tau) = (0.9, 4)$ for all methods [Hinton et al., 2015, Cho and Hariharan, 2019, Wang et al., 2020]. For CIFAR-100 and TinyImageNet, we used the value $(\alpha, \tau) = (0.9, 1)$ for all methods. We fix ODS step-size $\eta$ to $1/255$ across all settings. See Appendix B.1 for more details.

**Uncertainty metrics**  We measure uncertainty calibration performance with common metrics including Accuracy (ACC), Negative Log-Likelihood (NLL), Expected Calibration Error (ECE) and Brier Score (BS). We also report the Deep Ensemble Equivalent (DEE) score [Ashukha et al., 2020], which measures the effective number of models by comparing it to a full DE. For all methods, the metrics are computed at optimal temperatures obtained from temperature scaling [Guo et al., 2017], as suggested by Ashukha et al. [2020]. See Appendix B.2 for the definition of each metric.

### 5.2  Impact of ODS on diversities

In order to assess the effect of ODS perturbation on the diversities of DE teachers or student subnetworks, we draw *diversity plots* in Fig. 3. To draw a diversity plot, we first collect the outputs of target models evaluated on a specific dataset and binned the samples w.r.t. their confidence values (minimum confidence among $M$ models). For each bin, we computed average pairwise KL-divergence between class output probabilities of ensemble members. We also report the representative value (mean-KLD), which is defined as average KL-divergence values weighted by bin counts.

Fig. 3a shows the diversity plots of DE teachers on CIFAR-10. Intuitively, the samples predicted with high confidences usually come with low diversities, and this is indeed depicted in the plots. For the DE evaluated on the training set, the mean-KLD value is very small (the first box) compared to the one computed on the test set (the fourth box). While Gaussian perturbation does not affect the diversities (the second box), ODS drastically amplifies the diversities (the third box).

Fig. 3b shows the diversity plots of the BE students trained on CIFAR-10. Although evaluated at the test set, the BE distilled without any perturbation (the first box), the BE distilled with Gaussian perturbations (the second box), and the BE trained from scratch (the fourth box) exhibit very low diversities. On the other hand, the BE distilled with ODS perturbations show relatively high diversity, and as we will show later in Section 5.4, this indeed leads to improved performances.

In Section 3, we assumed that an ODS computed from a specific teacher can be transferred to other teachers. As an empirical justification, we draw the diversity plot using ODS computed from an

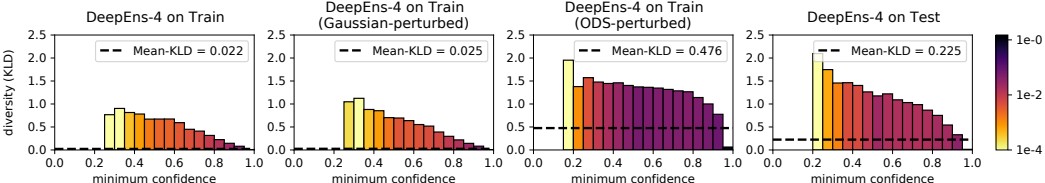

**(a)** Diversity plots of DE-4 teachers for ResNet-32 on train examples of CIFAR-10.

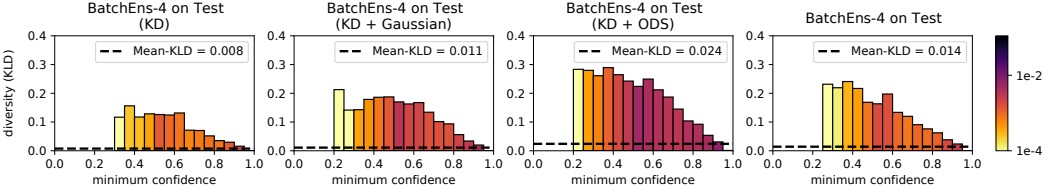

**(b)** Diversity plots of BE-4 students for ResNet-32 on test examples of CIFAR-10.

**Figure 3:** Bar colors denote the density of the bins, *i.e.*, ratio of samples belonging to the bins.

external teacher (a network trained in the same way as the teachers but not actually being used during distillation) in Fig. 6a. The mean-KLD values are indeed increased compared to the original DE or DE evaluated on examples perturbed with Gaussian noise, confirming the validity of our assumption.

### 5.3  ODS for Jacobian matching

**Effectiveness**   We empirically validate our conjecture that ODS perturbations are more effective in Jacobian matching. To this end, we computed the cosine similarities between the vectorized Jacobians of ResNet-32 teacher and student networks trained with vanilla KD. We computed the same quantity using student networks distilled with Gaussian perturbations and ODS perturbations. We draw Receiver Operating Characteristic (ROC) curves to compare the similarities of distilled students and the baseline trained from scratch. Specifically, we compare two distributions of the cosine similarities between the vectorized Jacobians of a teacher network and a student network: (1) when the student network is trained

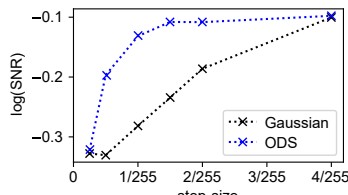

**Figure 4:** SNR values computed with Gaussian and ODS perturbations with varying step-sizes.

by KD, and (2) when the student network is trained from scratch without any guide from the teacher network. Results in Fig. 5 clearly show that the Jacobians of the student distilled with ODS better match the Jacobians, while the student distilled with Gaussian perturbations did not significantly improve upon the student distilled with vanilla KD.

Next, we compute the gradient of the term related to the Jacobian matching, *i.e.*,

$$G(\boldsymbol{x}, \boldsymbol{\varepsilon}) = \nabla_{\boldsymbol{\theta}}(\mathcal{L}_{\text{KD}}(\mathcal{S}(\boldsymbol{x} + \boldsymbol{\varepsilon}), \mathcal{T}(\boldsymbol{x} + \boldsymbol{\varepsilon})) - \mathcal{L}_{\text{KD}}(\mathcal{S}(\boldsymbol{x}), \mathcal{T}(\boldsymbol{x}))). \qquad (11)$$

Using both Gaussian perturbations and ODS perturbations, we collected multiple gradient samples and measured the Signal-to-Noise Ratio (SNR) of the gradient norms. Specifically, we computed $\text{SNR}(\boldsymbol{x}, \boldsymbol{\varepsilon}) = \|\mathbb{E}_{\boldsymbol{\varepsilon}}\left[\text{vec}(G(\boldsymbol{x}, \boldsymbol{\varepsilon}))\right]\|_2 / \sqrt{\|\text{Var}\left[\text{vec}(G(\boldsymbol{x}, \boldsymbol{\varepsilon}))\right]\|_2}$, where $\text{vec}(\cdot)$ is vectorization and $\text{Var}[\cdot]$ is element-wise variance. As shown in Fig. 4, the gradients computed with ODS perturbations exhibit higher SNR, indicating that learning with such gradients is more stable and efficient.

**Transferability**   As in Section 5.2, we performed an experiment to verify our assumption that ODS perturbations for a specific model are transferable to other models, for the purpose approximate Jacobian matching. We trained a BE student with ODS perturbations computed from an external teacher, and measured the cosine similarities between the student and teachers (*not* between the students and the external teacher). As one can see from Fig. 6b, the resulting student network copied the Jacobians of the teachers significantly better than the baselines.

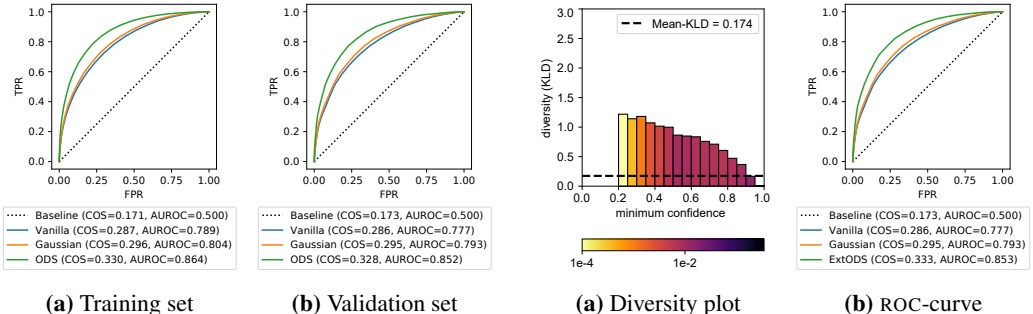

| **(a)** Training set | **(b)** Validation set | **(a)** Diversity plot | **(b)** ROC-curve |

**Figure 5:** ROC-curves with cosine similarities of Jacobians between a teacher and students. 'COS' denotes the cosine similarities between the teacher and student averaged over examples, and 'AUROC' denotes the area under the ROC-curves.

**Figure 6:** Transferability analysis using diversity plot and ROC-curve. Here, a BE student is trained with ODS perturbations computed from an external teacher, and it is denoted as 'ExtODS' in the ROC-curve for validation set.

**Table 1:** Knowledge distillation from DE-$M$ into BE-$M$ where $M$ denotes the size of ensembles: ACC and *calibrated metrics* including NLL, BS, ECE, and DEE. All values for ResNet-32 and WideResNet-28x10 are averaged over four and three experiments, respectively.

| | | **BatchEns « DeepEns (ResNet-32 on CIFAR-10)** | | | | |
|---|---|---|---|---|---|---|
| Method | # Params | ACC ($\uparrow$) | NLL ($\downarrow$) | BS ($\downarrow$) | ECE ($\downarrow$) | DEE ($\uparrow$) |
| $\mathcal{T}$ : DeepEns-4 | 1.86 M | 94.42 | 0.167 | 0.082 | 0.008 | - |
| $\mathcal{S}$ : BatchEns-4 | 0.47 M | $93.37_{\pm0.11}$ | $0.204_{\pm0.002}$ | $0.099_{\pm0.001}$ | $0.008_{\pm0.001}$ | $1.419_{\pm0.075}$ |
| + KD | | $93.98_{\pm0.20}$ | $0.188_{\pm0.003}$ | $0.091_{\pm0.002}$ | $0.009_{\pm0.002}$ | $2.019_{\pm0.174}$ |
| + KD + Gaussian | | $93.93_{\pm0.12}$ | $0.187_{\pm0.001}$ | $0.090_{\pm0.001}$ | $0.009_{\pm0.002}$ | $2.042_{\pm0.089}$ |
| + KD + ODS | | $93.89_{\pm0.10}$ | $0.181_{\pm0.002}$ | $0.090_{\pm0.001}$ | $\mathbf{0.006}_{\pm0.001}$ | $2.486_{\pm0.164}$ |
| + KD + ConfODS | | $\mathbf{94.01}_{\pm0.19}$ | $\mathbf{0.180}_{\pm0.001}$ | $\mathbf{0.089}_{\pm0.001}$ | $0.007_{\pm0.001}$ | $\mathbf{2.524}_{\pm0.080}$ |
| $\mathcal{T}$ : DeepEns-8 | 3.71 M | 94.78 | 0.157 | 0.077 | 0.005 | - |
| $\mathcal{S}$ : BatchEns-8 | 0.48 M | $93.47_{\pm0.14}$ | $0.202_{\pm0.005}$ | $0.098_{\pm0.002}$ | $\mathbf{0.006}_{\pm0.002}$ | $1.494_{\pm0.156}$ |
| + KD | | $94.15_{\pm0.13}$ | $0.182_{\pm0.001}$ | $0.088_{\pm0.001}$ | $0.010_{\pm0.001}$ | $2.391_{\pm0.113}$ |
| + KD + Gaussian | | $94.09_{\pm0.08}$ | $0.184_{\pm0.002}$ | $0.089_{\pm0.001}$ | $0.010_{\pm0.002}$ | $2.206_{\pm0.175}$ |
| + KD + ODS | | $94.13_{\pm0.08}$ | $0.175_{\pm0.003}$ | $\mathbf{0.086}_{\pm0.001}$ | $\mathbf{0.006}_{\pm0.002}$ | $2.991_{\pm0.322}$ |
| + KD + ConfODS | | $\mathbf{94.18}_{\pm0.12}$ | $\mathbf{0.174}_{\pm0.002}$ | $\mathbf{0.086}_{\pm0.001}$ | $0.007_{\pm0.001}$ | $\mathbf{3.064}_{\pm0.246}$ |

| | | **BatchEns « DeepEns (WRN-28x10 on CIFAR-100)** | | | | |
|---|---|---|---|---|---|---|
| Method | # Params | ACC ($\uparrow$) | NLL ($\downarrow$) | BS ($\downarrow$) | ECE ($\downarrow$) | DEE ($\uparrow$) |
| $\mathcal{T}$ : DeepEns-4 | 146.15 M | 82.52 | 0.661 | 0.247 | 0.022 | - |
| $\mathcal{S}$ : BatchEns-4 | 36.62 M | $80.34_{\pm0.08}$ | $0.755_{\pm0.007}$ | $0.280_{\pm0.002}$ | $0.027_{\pm0.001}$ | $1.449_{\pm0.085}$ |
| + KD | | $80.51_{\pm0.22}$ | $0.744_{\pm0.003}$ | $0.274_{\pm0.001}$ | $\mathbf{0.021}_{\pm0.003}$ | $1.582_{\pm0.035}$ |
| + KD + Gaussian | | $80.39_{\pm0.12}$ | $0.761_{\pm0.006}$ | $0.277_{\pm0.000}$ | $0.022_{\pm0.003}$ | $1.379_{\pm0.072}$ |
| + KD + ODS | | $\mathbf{81.88}_{\pm0.32}$ | $0.674_{\pm0.016}$ | $0.257_{\pm0.006}$ | $0.026_{\pm0.003}$ | $3.303_{\pm0.769}$ |
| + KD + ConfODS | | $81.85_{\pm0.32}$ | $\mathbf{0.672}_{\pm0.010}$ | $\mathbf{0.256}_{\pm0.003}$ | $0.024_{\pm0.000}$ | $\mathbf{3.333}_{\pm0.491}$ |

## 5.4  Main results: image classification tasks

We validate whether the enhanced diversity of student networks is helpful for actual prediction. We compared our methods against three baselines: BE trained from scratch [Wen et al., 2020], BE distilled without perturbation [Mariet et al., 2020], and BE distilled with Gaussian noises.

The results are presented in Table 1. For both datasets, BE distilled with ODS or ConfODS significantly outperforms baselines in terms of every metric we measured. The improvement is much more significant for the larger ensembles ($M = 8$ for CIFAR-10) and the larger dataset (CIFAR-100). We note that using our diversification strategy improves performance both in terms of raw accuracy and uncertainty metrics. An especially noteworthy result on CIFAR-100 is that BE distilled with ODS achieved a DEE score over three. This is quite remarkable considering the BE model has far fewer parameters than the full DE which, by definition, has an expected DEE of four. Moreover, further uncertainty results in the Appendix A.1 show that our approach is superior with respect to both predictive uncertainty for out-of-distribution examples and calibration on corrupted datasets.

**Table 2:** Cross-architecture knowledge distillation for a model compression on CIFAR-100 and TinyImageNet: BE-4 students with smaller networks (*i.e.*, WideResNet-28x2 and WideResNet-28x5), learn from the DE-4 teacher which consists of larger networks (*i.e.*, WideResNet-28x10). All values for CIFAR-100 and TinyImageNet are averaged over three and one experiments, respectively.

| | | BatchEns-4 « DeepEns-4 (WRN-28x2 on CIFAR-100) | | | |
|---|---|---|---|---|---|
| Method | # Params | ACC ($\uparrow$) | NLL ($\downarrow$) | BS ($\downarrow$) | ECE ($\downarrow$) |
| $\mathcal{T}$ : DeepEns-4 | 146.15 M | 82.52 | 0.676 | 0.250 | 0.035 |
| $\mathcal{S}$ : BatchEns-4 | 1.50 M | $75.17_{\pm 0.27}$ | $1.245_{\pm 0.024}$ | $0.383_{\pm 0.004}$ | $0.141_{\pm 0.004}$ |
| + KD | | $75.19_{\pm 0.36}$ | $1.207_{\pm 0.021}$ | $0.377_{\pm 0.007}$ | $0.136_{\pm 0.005}$ |
| + KD + Gaussian | | $74.50_{\pm 0.17}$ | $1.247_{\pm 0.012}$ | $0.389_{\pm 0.004}$ | $0.137_{\pm 0.003}$ |
| + KD + ODS | | $\mathbf{76.03}_{\pm 0.22}$ | $\mathbf{0.899}_{\pm 0.013}$ | $\mathbf{0.333}_{\pm 0.003}$ | $\mathbf{0.027}_{\pm 0.002}$ |
| + KD + ConfODS | | $76.01_{\pm 0.16}$ | $0.901_{\pm 0.006}$ | $0.334_{\pm 0.002}$ | $0.028_{\pm 0.004}$ |

| | | BatchEns-4 « DeepEns-4 (WRN-28x5 on CIFAR-100) | | | |
|---|---|---|---|---|---|
| Method | # Params | ACC ($\uparrow$) | NLL ($\downarrow$) | BS ($\downarrow$) | ECE ($\downarrow$) |
| $\mathcal{T}$ : DeepEns-4 | 146.15 M | 82.52 | 0.661 | 0.247 | 0.022 |
| $\mathcal{S}$ : BatchEns-4 | 9.20 M | $78.75_{\pm 0.11}$ | $0.801_{\pm 0.012}$ | $0.297_{\pm 0.003}$ | $0.021_{\pm 0.002}$ |
| + KD | | $78.89_{\pm 0.10}$ | $0.804_{\pm 0.012}$ | $0.296_{\pm 0.003}$ | $0.022_{\pm 0.001}$ |
| + KD + Gaussian | | $78.80_{\pm 0.41}$ | $0.815_{\pm 0.009}$ | $0.297_{\pm 0.005}$ | $\mathbf{0.020}_{\pm 0.002}$ |
| + KD + ODS | | $80.24_{\pm 0.05}$ | $0.742_{\pm 0.008}$ | $0.279_{\pm 0.002}$ | $0.028_{\pm 0.004}$ |
| + KD + ConfODS | | $\mathbf{80.62}_{\pm 0.25}$ | $\mathbf{0.733}_{\pm 0.007}$ | $\mathbf{0.275}_{\pm 0.003}$ | $0.027_{\pm 0.001}$ |

| | | BatchEns-4 « DeepEns-4 (WRN-28x5 on TinyImageNet) | | | |
|---|---|---|---|---|---|
| Method | # Params | ACC ($\uparrow$) | NLL ($\downarrow$) | BS ($\downarrow$) | ECE ($\downarrow$) |
| $\mathcal{T}$ : DeepEns-4 | 146.40 M | 69.90 | 1.242 | 0.403 | 0.016 |
| $\mathcal{S}$ : BatchEns-4 | 9.23 M | 64.86 | 1.455 | 0.464 | $\mathbf{0.022}$ |
| + KD | | 65.86 | 1.432 | $\mathbf{0.456}$ | 0.022 |
| + KD + Gaussian | | 65.72 | 1.446 | 0.457 | 0.022 |
| + KD + ODS | | $\mathbf{65.98}$ | $\mathbf{1.408}$ | $\mathbf{0.456}$ | 0.022 |

One great benefit of the KD framework is that it can transfer knowledge to networks having different architectures, greatly reducing the number of parameters required to achieve a certain level of performance. Using CIFAR-100 and TinyImageNet datasets, we compared our method to baselines with student networks having smaller architecture. The results displayed in Table 2 show that KD with ODS or ConfODS outperforms the baselines by a wide margin.

## 6   Conclusion

In this paper, we proposed a simple yet effective training scheme for distilling knowledge from an ensemble of teachers. The key idea of our approach is to perturb training data in a proper way to expose student networks to the diverse outputs produced by ensemble teachers. We employed ODS to find such diversifying perturbation. Our distillation scheme with ODS perturbation can be interpreted as a sample-efficient approximate Jacobian matching procedure. We empirically verified that our method enhances the diversities of students, and that such enhanced diversities help to reduce the performance gap between student and teacher models in KD. An interesting future work would be to rigorously analyze the Jacobian matching aspect of our method and to provide theoretical justification and a better distillation method to harness it. Also, in this paper, we constrained ourselves to BE students, and applying our framework to other student network architectures such as Bayesian neural networks would be interesting.

**Limitations**   Although we have shown promising results, our method still has several limitations. Due to its nature as a KD framework, the need to train $M$ teachers involves a considerable computational cost. Also, our method cannot improve the student beyond its capacity. For instance, in Table 2, WideResNet-28x2 distilled from WideResNet-28x10 does not greatly benefit from our method. Arguably the most critical limitation is that our method is a heuristic and lacks any theoretical guarantee. In that sense, as mentioned above, rigorously analyzing the effect of ODS perturbation will be important future research.

## Acknowledgement

This work was partly supported by Institute of Information & communications Technology Planning & Evaluation (IITP) grant funded by the Korea government (MSIT) (No.2019-0-00075, Artificial Intelligence Graduate School Program(KAIST), No. 2021-0-02068, Artificial Intelligence Innovation Hub), National Research Foundation of Korea (NRF) funded by the Ministry of Education (NRF-2021R1F1A1061655, NRF-2021M3E5D9025030), and Samsung Electronics Co., Ltd (IO201214-08176-01).

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
