# A Additional Experiments

## A.1 Further evaluation on uncertainty

**Predictive uncertainty for out-of-distribution examples** For reliable deployment in real-world decision making systems, deep neural networks should not make over-confident predictions when inputs are far from the training data. Following Lakshminarayanan et al. [2017], we evaluate the entropy of predictions on out-of-distribution examples from unseen classes to quantify the quality of the predictive uncertainty. We use the test split of the SVHN dataset as out-of-distribution data; it consists of 26,032 digit images with the same image size as CIFAR-10/100[2]. As shown in Fig. 7 and Fig. 8, while DE exhibits the highest out-of-distribution entropy, our models show comparable out-of-distribution entropy. Especially, for WideResNet-28x10 models trained on CIFAR-100, our method almost matches the predictive uncertainty of DE teachers (*i.e.*, 2.072 versus 2.093).

**Calibration on corrupted datasets** We also evaluate predictive uncertainty on the corrupted versions of CIFAR datasets; it consists of 15 types of corruptions on the original test examples of CIFAR datasets where each corruption type has five intensities [Hendrycks and Dietterich, 2019]. In order to compare models in terms of calibration on corrupted datasets, we draw box-and-whisker plots for NLL and ECE measuring calibration across corruption types and intensities in Fig. 9. It clearly shows that BE students distilled with ODS are better calibrated than the baselines. We also include standard metrics averaged over all corruption types and intensities in Table 3. A remarkable result on CIFAR-100 is that BE students with ODS achieve comparable calibration to DE teachers both in Fig. 9 and Table 3.

## A.2 Adversarial perturbation instead of ODS

The ODS perturbation is very similar to the adversarial perturbation; if we replace the uniform vector with the one-hot class labels in Eq. (3), we get an adversarial perturbation. The difference is that the adversarial perturbation is meant to worsen the predictive performance by design because it takes a step toward the directions increasing the classification loss. We empirically found that the perturbations just increasing diversity without maintaining prediction accuracy can actually harm the performance of student models (this is also related to the performance gain of ConfODS). In specific, we tested the KD with adversarial perturbations (*i.e.*, non-targeted attack) instead of ODS perturbations using ResNet-32 on CIFAR-10. This achieved ACC of $93.32 \pm 0.22$ and NLL of $0.194 \pm 0.003$, which is much worse than vanilla KD with no perturbation presented in Table 1.

## A.3 Changing student's architecture

We performed additional experiments using the Multi-Input Multi-Output (MIMO) architecture [Havasi et al., 2021]. The biggest difference between MIMO and BE is in how subnetworks are constructed. In short, unlike BE, MIMO constructs its subnetworks without any explicit weight parameterization. Even though MIMO constructs its multiple subnetworks implicitly, it can still be trained using our method. We applied KD from DE-3 teachers to MIMO-3 students with Gaussian and ODS perturbations using WideResNet-28x10 on CIFAR-100, and the results are summarized in Table 4. Again, ours significantly improved the performance compared to the vanilla KD or KD with Gaussian perturbation. This result demonstrates that our proposed method is broadly beneficial independently of multi-network architecture and is not specific to BE.

## A.4 Comparison with Ensemble Distribution Distillation (END$^2$)

To address the issue of distilling diversity more thoroughly, we performed additional experiments comparing to the recently proposed Ensemble Distribution Distillation (END$^2$) [Malinin et al., 2020], which considers the distribution of DE teacher predictions. Here, we adopted the publicly available PyTorch implementation of END$^2$, and compared it with our method using ResNet-32 on CIFAR-10[3]. Quantitative results in Table 5 show that ours significantly outperforms END$^2$ in terms of both accuracy and uncertainty estimation metrics.

---

[2] http://ufldl.stanford.edu/housenumbers/
[3] https://github.com/lennelov/endd-reproduce/tree/d61d298b52c4338e07d7cd4a3fdc65f1de1bcbf1

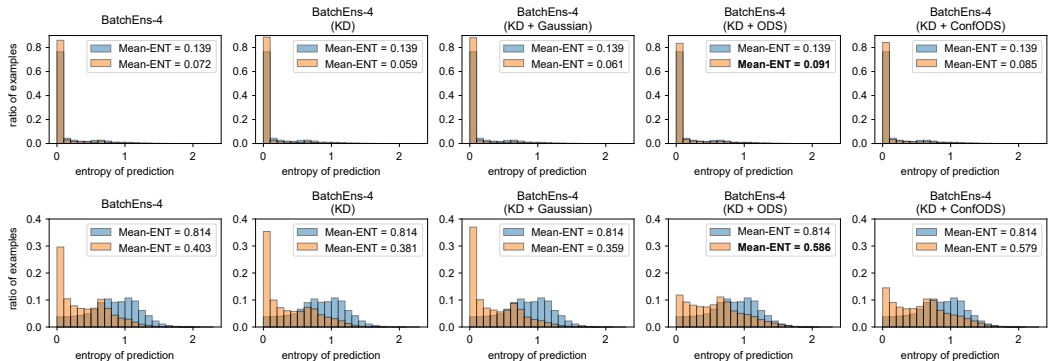

**Figure 7:** Histogram of the predictive entropy of ResNet-32 models on test examples from known classes, *i.e.*, CIFAR-10 (top row), and unknown classes, *i.e.*, SVHN (bottom row). The histograms with mean entropies of 0.139 on CIFAR-10 and 0.814 on SVHN denote DeepEns-4 teacher.

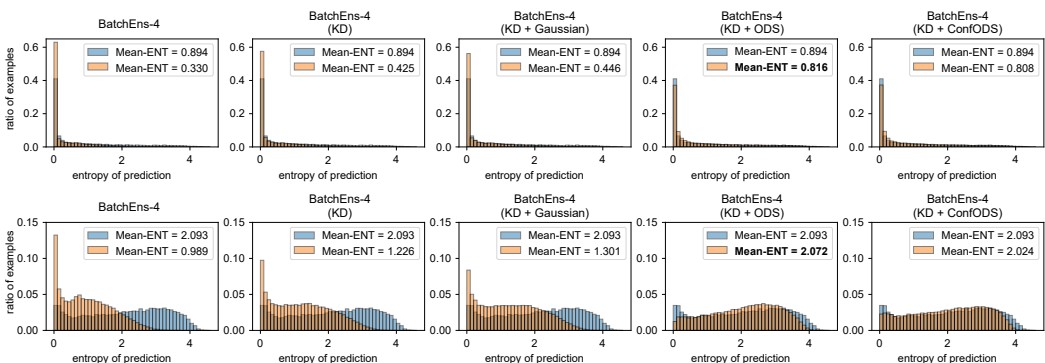

**Figure 8:** Histogram of the predictive entropy of WideResNet-28x10 models on test examples from known classes, *i.e.*, CIFAR-100 (top row), and unknown classes, *i.e.*, SVHN (bottom row). The histograms with mean entropies of 0.894 on CIFAR-100 and 2.093 on SVHN denote DeepEns-4 teacher.

## B  Experimental Details

Code is available at https://github.com/cs-giung/giung2/tree/main/projects/Diversity-Matters.

### B.1  Training

**CIFAR-10/100**   CIFAR-10/100 consists of a train set of 50,000 images and a test set of 10,000 images from 10/100 classes, with images size of $32 \times 32 \times 3$[4]. All models are trained on the first 45k examples of the train split of CIFAR datasets and the last 5k examples of the train split are used as the validation split. We follow the standard data augmentation policy [He et al., 2016] which consists of random cropping of 32 pixels with a padding of 4 pixels and random horizontal flipping.

**TinyImageNet**   TinyImageNet is a subset of ImageNet dataset consisting of 100,000 images from 200 classes with images resized to $64 \times 64 \times 3$[5]. Since the labels of the official test set are not publicly available, we use the official validation set consisting of 10,000 images as a test set for experiments. All models are trained on the first 450 examples for each class and the last 50 examples for each class are used as validation examples, *i.e.*, train and validation splits consist of 90k and 10k examples, respectively. We use a data augmentation which consists of random cropping of 64 pixels with a padding of 8 pixels and random horizontal flipping.

---

[4]https://www.cs.toronto.edu/ ~kriz/cifar.html
[5]http://cs231n.stanford.edu/tiny-imagenet-200.zip

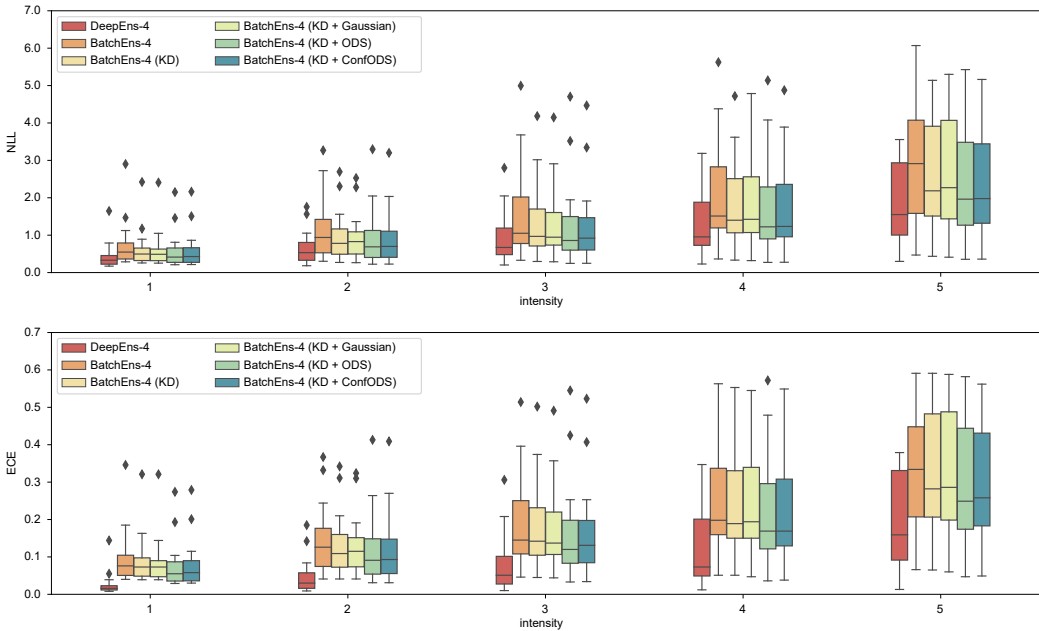

**(a)** ResNet-32 models on CIFAR-10-C.

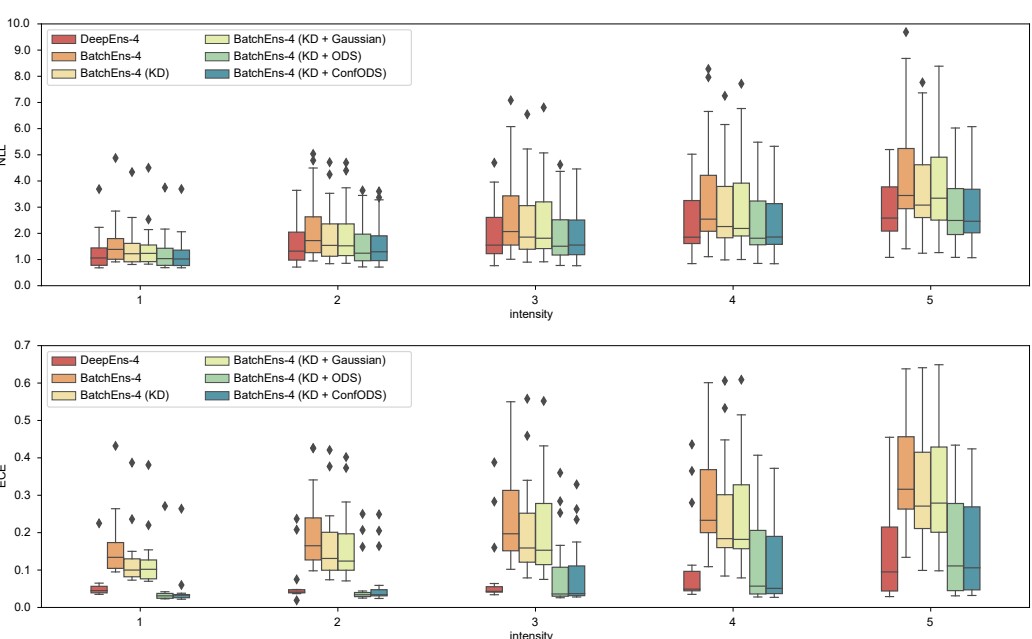

**(b)** WideResNet-28x10 models on CIFAR-100-C.

**Figure 9:** Calibration on CIFAR-10-C and CIFAR-100-C: box-and-whisker plot shows medians of NLL and ECE across corruption types for five levels of intensity.

**Table 3:** Evaluation results on CIFAR-10-C and CIFAR-100-C: ACC and *standard metrics* including NLL, BS, and ECE. All values for ResNet-32 and WideResNet-28x10 are averaged over four and three experiments, respectively.

| Method | # Params | **BatchEns-4 « DeepEns-4 (ResNet-32 on CIFAR-10)** | | | |
|---|---|---|---|---|---|
| | | ACC ($\uparrow$) | NLL ($\downarrow$) | BS ($\downarrow$) | ECE ($\downarrow$) |
| $\mathcal{T}$ : DeepEns-4 | 1.86 M | 73.51 | 1.036 | 0.380 | 0.093 |
| $\mathcal{S}$ : BatchEns-4 | 0.47 M | $70.39_{\pm 0.65}$ | $1.723_{\pm 0.086}$ | $0.485_{\pm 0.012}$ | $0.204_{\pm 0.006}$ |
| + KD | | $72.68_{\pm 0.49}$ | $1.485_{\pm 0.051}$ | $0.455_{\pm 0.010}$ | $0.196_{\pm 0.006}$ |
| + KD + Gaussian | | $\mathbf{72.71}_{\pm 0.38}$ | $1.487_{\pm 0.026}$ | $\mathbf{0.453}_{\pm 0.008}$ | $0.194_{\pm 0.004}$ |
| + KD + ODS | | $70.50_{\pm 0.46}$ | $1.406_{\pm 0.044}$ | $0.462_{\pm 0.009}$ | $\mathbf{0.178}_{\pm 0.006}$ |
| + KD + ConfODS | | $70.80_{\pm 0.36}$ | $\mathbf{1.404}_{\pm 0.038}$ | $0.459_{\pm 0.008}$ | $0.180_{\pm 0.005}$ |

| Method | # Params | **BatchEns-4 « DeepEns-4 (WRN28x10 on CIFAR-100)** | | | |
|---|---|---|---|---|---|
| | | ACC ($\uparrow$) | NLL ($\downarrow$) | BS ($\downarrow$) | ECE ($\downarrow$) |
| $\mathcal{T}$ : DeepEns-4 | 146.15 M | 53.86 | 2.091 | 0.609 | 0.096 |
| $\mathcal{S}$ : BatchEns-4 | 36.62 M | $52.36_{\pm 0.44}$ | $2.953_{\pm 0.124}$ | $0.708_{\pm 0.009}$ | $0.255_{\pm 0.006}$ |
| + KD | | $53.00_{\pm 0.33}$ | $2.585_{\pm 0.023}$ | $0.677_{\pm 0.005}$ | $0.216_{\pm 0.003}$ |
| + KD + Gaussian | | $52.81_{\pm 0.65}$ | $2.654_{\pm 0.009}$ | $0.682_{\pm 0.004}$ | $0.218_{\pm 0.001}$ |
| + KD + ODS | | $53.87_{\pm 1.00}$ | $2.109_{\pm 0.073}$ | $0.614_{\pm 0.013}$ | $0.104_{\pm 0.009}$ |
| + KD + ConfODS | | $\mathbf{54.19}_{\pm 0.75}$ | $\mathbf{2.083}_{\pm 0.073}$ | $\mathbf{0.610}_{\pm 0.015}$ | $\mathbf{0.101}_{\pm 0.011}$ |

**Table 4:** Knowledge distillation from DE-3 into MIMO-3: ACC, *standard metrics* and *calibrated metrics*. All values are measured one time.

| Method | # Params | **Standard Metrics** | | | | **Calibrated Metrics** | | |
|---|---|---|---|---|---|---|---|---|
| | | ACC ($\uparrow$) | NLL ($\downarrow$) | BS ($\downarrow$) | ECE ($\downarrow$) | NLL ($\downarrow$) | BS ($\downarrow$) | ECE ($\downarrow$) |
| $\mathcal{T}$ : DeepEns-3 | 109.61 M | 82.43 | 0.690 | 0.253 | 0.036 | 0.677 | 0.251 | 0.025 |
| $\mathcal{S}$ : MIMO-3 | 36.67 M | 80.63 | 0.720 | 0.271 | 0.027 | 0.716 | 0.270 | $\mathbf{0.014}$ |
| + KD | | 80.75 | 0.723 | 0.271 | 0.029 | 0.720 | 0.271 | 0.021 |
| + KD + Gaussian | | 80.69 | 0.726 | 0.272 | $\mathbf{0.024}$ | 0.723 | 0.271 | 0.016 |
| + KD + ODS | | $\mathbf{81.17}$ | $\mathbf{0.715}$ | $\mathbf{0.270}$ | 0.037 | $\mathbf{0.692}$ | $\mathbf{0.267}$ | 0.020 |

**Learning rate schedules** We use the following learning rate scheduling suggested in Ashukha et al. [2020] for all experiments.

1. For the first five epochs, linearly increase the learning rate from $0.01 \times$ `base_lr` to `base_lr`.

2. Until $0.5 \times$ `total_epochs`, keep the learning rate as `base_lr`.

3. From $0.5 \times$ `total_epochs` to $0.9 \times$ `total_epochs`, linearly decay the learning rate to $0.01 \times$ `base_lr`.

4. From $0.9 \times$ `total_epochs` to `total_epochs`, keep the learning rate as $0.01 \times$ `base_lr`, and save the model with the best validation accuracy.

**Optimization** We use SGD optimizer with momentum $0.9$ for all experiments. Specifically,

- We use the optimizer with batch size $512$, `base_lr` $0.4$ and weight decay parameter $4 \times 10^{-4}$ to train ResNet-32 on CIFAR-10. The `total_epochs` is set to $200$.

- We use the optimizer with batch size $256$, `base_lr` $0.2$ and weight decay parameter $5 \times 10^{-4}$ to train WideResNet-28x10 on CIFAR-100. The `total_epochs` is set to $300$.

- We use the optimizer with batch size $512$, `base_lr` $0.4$ and weieght decay parameter $5 \times 10^{-4}$ to train WideResNet-28x10 on TinyImageNet. The `total_epochs` is set to $300$.

**Table 5:** Comparison between END$^2$ and ours (*i.e.*, ConfODS) distilling from DE-$M$ teachers where $M$ denotes the size of ensembles using ResNet-32 on CIFAR-10: ACC and *calibrated metrics* including NLL, BS, and ECE. All values are measured one time.

| | END$^2$ | | | | Ours (ConfODS) | | | |
|---|---|---|---|---|---|---|---|---|
| Teacher | ACC ($\uparrow$) | NLL ($\downarrow$) | BS ($\downarrow$) | ECE ($\downarrow$) | ACC ($\uparrow$) | NLL ($\downarrow$) | BS ($\downarrow$) | ECE ($\downarrow$) |
| $\mathcal{T}$ : DeepEns-4 | 92.10 | 0.260 | 0.121 | 0.017 | 94.01 | 0.180 | 0.093 | **0.007** |
| $\mathcal{T}$ : DeepEns-8 | 93.11 | 0.225 | 0.104 | 0.013 | **94.18** | **0.174** | **0.091** | **0.007** |
| $\mathcal{T}$ : DeepEns-16 | 93.21 | 0.221 | 0.104 | 0.016 | - | - | - | - |
| $\mathcal{T}$ : DeepEns-32 | 93.49 | 0.218 | 0.102 | 0.014 | - | - | - | - |

To obtain DE-$M$ teacher models, we repeat the described training procedure $M$ times with different random seeds. For BE-$M$ models, we follow the latest official implementation[6]. In specific, we train all the subnetworks with the same mini-batches; it can be done by repeating the training of a single batch $M$ times during training. As a result,

- We use the optimizer with batch size $128 \times 4$, `base_lr` 0.1 and weight decay parameter $4 \times 10^{-4}$ to train BE-4 for ResNet-32 on CIFAR-10. The `total_epochs` is set to 250, as suggested in Wen et al. [2020].
- We use the optimizer with batch size $64 \times 8$, `base_lr` 0.05 and weight decay parameter $4 \times 10^{-4}$ to train BE-8 for ResNet-32 on CIFAR-10. The `total_epochs` is set to 250.
- We use the optimizer with batch size $64 \times 4$, `base_lr` 0.05 and weight decay parameter $5 \times 10^{-4}$ to train BE-4 for WideResNet models on CIFAR-100 and TinyImageNet.

**Hyperparameters for knowledge distillation** We searched for several $(\alpha, \tau)$ pairs via grid search to run our experiments with the best hyperparameters. As shown in Table 6, we did not find a significant effect of different hyperparameter settings on performance for ResNet-32 on CIFAR-10. We therefore decided to stay consistent with the convention, *i.e.*, $\alpha = 0.9$ and $\tau = 4$. However, as shown in Table 7, we empirically found that $\tau = 4$ is not suitable to perform knowledge distillation for WideResNet-28x10 on CIFAR-100, and decided to use $\tau = 1$ which achieved the best NLL across temperature values. From this, we decided to use $\alpha = 0.9$ and $\tau = 4$ for WideResNet models on CIFAR-100 and TinyImageNet.

## B.2 Evaluation

For further metric descriptions, we denote a neural network as $\mathcal{F}(\boldsymbol{x}) : \mathbb{R}^D \to [0, 1]^K$; $\mathcal{F}$ outputs class probabilities over $K$ classes, and we denote the logits before softmax as $\hat{\mathcal{F}}(\boldsymbol{x})$.

**Standard metrics** For a model $\mathcal{F}$, standard metrics including accuracy, negative log-likelihood, Brier score and expected calibration error are defined as follows:

- Accuracy (ACC):

$$\text{ACC}(\mathcal{F}) = \Pr_{(\boldsymbol{x},\boldsymbol{y})\in\mathcal{D}} \left[ \arg\max_{k\in\{1...K\}} \left\{ \mathcal{F}^{(k)}(\boldsymbol{x}) \right\} = \arg\max_{k\in\{1...K\}} \left\{ \boldsymbol{y}^{(k)} \right\} \right]. \quad (12)$$

- Negative Log-Likelihood (NLL):

$$\text{NLL}(\mathcal{F}) = \mathbb{E}_{(\boldsymbol{x},\boldsymbol{y})\in\mathcal{D}} \left[ -\sum_{k=1}^{K} \boldsymbol{y}^{(k)} \log \mathcal{F}^{(k)}(\boldsymbol{x}) \right]. \quad (13)$$

- Brier Score (BS):

$$\text{BS}(\mathcal{F}) = \mathbb{E}_{(\boldsymbol{x},\boldsymbol{y})\in\mathcal{D}} \left[ \frac{1}{K} \sum_{k=1}^{K} \left( \mathcal{F}^{(k)}(\boldsymbol{x}) - \boldsymbol{y}^{(k)} \right)^2 \right]. \quad (14)$$

---

[6]https://github.com/google/uncertainty-baselines/tree/ffa818a665655c37e921b411512191ad260cfb47

**Table 6:** Validation ACC and calibrated NLL for different values of hyperparameters $\alpha$ and $\tau$ for knowledge distillation from DE-4 into BE-4 of ResNet-32 on CIFAR-10. All values are measured four times.

| | ACC@Valid | | | NLL@Valid | | |
|---|---|---|---|---|---|---|
| | $\alpha = 0.8$ | $\alpha = 0.9$ | $\alpha = 1.0$ | $\alpha = 0.8$ | $\alpha = 0.9$ | $\alpha = 1.0$ |
| $\tau = 2$ | $94.67_{\pm 0.18}$ | $94.68_{\pm 0.17}$ | $94.59_{\pm 0.16}$ | $0.169_{\pm 0.005}$ | $0.170_{\pm 0.003}$ | $0.171_{\pm 0.003}$ |
| $\tau = 3$ | $94.64_{\pm 0.16}$ | $94.59_{\pm 0.11}$ | $94.57_{\pm 0.11}$ | $0.172_{\pm 0.003}$ | $0.173_{\pm 0.003}$ | $0.174_{\pm 0.006}$ |
| $\tau = 4$ | $94.48_{\pm 0.20}$ | $94.60_{\pm 0.14}$ | $94.58_{\pm 0.15}$ | $0.172_{\pm 0.002}$ | $0.174_{\pm 0.003}$ | $0.176_{\pm 0.001}$ |
| $\tau = 5$ | $94.50_{\pm 0.22}$ | $94.46_{\pm 0.15}$ | $94.45_{\pm 0.20}$ | $0.172_{\pm 0.001}$ | $0.175_{\pm 0.004}$ | $0.176_{\pm 0.005}$ |

**Table 7:** Validation ACC and calibrated NLL for different values of hyperparameters $\tau$ for knowledge distillation from DE-4 into BE-4 of WideResNet-28x10 on CIFAR-100 where $\alpha$ is fixed to 0.9. All values are measured one time.

| | $\tau = 1$ | $\tau = 2$ | $\tau = 3$ | $\tau = 4$ | $\tau = 5$ |
|---|---|---|---|---|---|
| ACC@Valid | 80.29 | 80.32 | 80.08 | 80.52 | 80.98 |
| NLL@Valid | 0.757 | 0.835 | 0.831 | 0.827 | 0.818 |

- Expected Calibration Error (ECE):

$$\text{ECE}(\mathcal{F}) = \sum_{l=1}^{L} \frac{|B_l|}{N} \left| \text{acc}(B_l) - \text{conf}(B_l) \right|, \tag{15}$$

where $N$ is the total number of examples; $|B_l|$ denotes the number of predictions in $l$th bin; $\text{acc}(B_l)$ and $\text{conf}(B_l)$ respectively denote the mean accuracy and mean confidence of predictions in $l$th bin.

**Calibrated metrics** For a model $\mathcal{F}$, we first compute the optimal temperature which minimizes a negative log-likelihood over validation set $\mathcal{D}_{\text{val}}$ as

$$\tau^* = \arg\min_{\tau > 0} \left\{ \mathbb{E}_{(\boldsymbol{x},\boldsymbol{y}) \in \mathcal{D}_{\text{val}}} \left[ -\sum_{k=1}^{K} \boldsymbol{y}^{(k)} \log \text{softmax}\left( \hat{\mathcal{F}}^{(k)}(\boldsymbol{x})/\tau \right) \right] \right\}, \tag{16}$$

and then, compute evaluation metrics with temperature scaled outputs:

$$\mathcal{F}^{(k)}(\boldsymbol{x} \; ; \; \tau^*) \leftarrow \text{softmax}\left( \hat{\mathcal{F}}^{(k)}(\boldsymbol{x})/\tau^* \right), \quad \text{where } k \in \{1, \dots, K\}. \tag{17}$$

**Evaluation for ensemble models** DE and BE construct a prediction by averaging the outputs of ensemble members (subnetworks) as follows:

$$\mathcal{F}^{(k)}(\boldsymbol{x}) \leftarrow \sum_{m=1}^{M} \mathcal{F}_m^{(k)}(\boldsymbol{x})/M, \tag{18}$$

where $k \in \{1...K\}$ and $\mathcal{F}_m$ denotes $m$th ensemble member. Therefore, we define the logits of the ensemble predictions as

$$\hat{\mathcal{F}}^{(k)}(\boldsymbol{x}) \leftarrow \log \left( \sum_{m=1}^{M} \mathcal{F}_m^{(k)}(\boldsymbol{x})/M \right), \quad \text{where } k \in \{1...K\}, \tag{19}$$

with which both standard and calibrated metrics can be computed. Also, we also compute the Deep Ensemble Equivalent (DEE) score [Ashukha et al., 2020]; for a model $\mathcal{S}$, DEE score is defined as

$$\text{DEE}(\mathcal{S}) = \min\{\ell \geq 0 \,|\, \text{NLL}(\text{Ensemble of } \ell \text{ models}) \leq \text{NLL}(\mathcal{S})\}. \tag{20}$$

That is, DEE computes the minimum number of independent ensemble members needed to achieve the same performance as a given model $\mathcal{S}$. The NLL values of ensembles with non-integer $\ell$ values are obtained by linear interpolation. Refer to Table 8 and Table 9 for evaluation results with standard metrics, and Table 1 and Table 2 for evaluation results with calibrated metrics.

**Table 8:** Knowledge distillation from DE-$M$ into BE-$M$ where $M$ denotes the size of ensembles: ACC and *standard metrics* including NLL, BS, ECE, and DEE. Refer to Table 1 for the results with calibrated metrics.

| Method | # Params | ACC | NLL | BS | ECE | DEE |
|---|---|---|---|---|---|---|
| **BatchEns « DeepEns (ResNet-32 on CIFAR-10)** | | | | | | |
| $\mathcal{T}$ : DeepEns-4 | 1.86 M | 94.42 | 0.170 | 0.081 | 0.008 | - |
| $\mathcal{S}$ : BatchEns-4 | 0.47 M | $93.37_{\pm 0.11}$ | $0.282_{\pm 0.003}$ | $0.107_{\pm 0.001}$ | $0.039_{\pm 0.001}$ | $0.994_{\pm 0.001}$ |
| + KD | | $93.98_{\pm 0.20}$ | $0.252_{\pm 0.005}$ | $0.099_{\pm 0.002}$ | $0.038_{\pm 0.001}$ | $1.261_{\pm 0.074}$ |
| + KD + Gaussian | | $93.93_{\pm 0.12}$ | $0.248_{\pm 0.004}$ | $0.099_{\pm 0.002}$ | $0.038_{\pm 0.002}$ | $1.316_{\pm 0.062}$ |
| + KD + ODS | | $93.89_{\pm 0.10}$ | $\mathbf{0.206}_{\pm 0.004}$ | $0.094_{\pm 0.001}$ | $\mathbf{0.028}_{\pm 0.001}$ | $\mathbf{1.938}_{\pm 0.067}$ |
| + KD + ConfODS | | $\mathbf{94.01}_{\pm 0.19}$ | $0.211_{\pm 0.002}$ | $\mathbf{0.093}_{\pm 0.001}$ | $0.029_{\pm 0.001}$ | $1.865_{\pm 0.036}$ |
| $\mathcal{T}$ : DeepEns-8 | 3.71 M | 94.78 | 0.157 | 0.077 | 0.004 | - |
| $\mathcal{S}$ : BatchEns-8 | 0.48 M | $93.47_{\pm 0.14}$ | $0.263_{\pm 0.011}$ | $0.104_{\pm 0.003}$ | $0.035_{\pm 0.002}$ | $1.123_{\pm 0.123}$ |
| + KD | | $94.15_{\pm 0.13}$ | $0.243_{\pm 0.003}$ | $0.095_{\pm 0.001}$ | $0.035_{\pm 0.002}$ | $1.394_{\pm 0.039}$ |
| + KD + Gaussian | | $94.09_{\pm 0.08}$ | $0.245_{\pm 0.003}$ | $0.097_{\pm 0.001}$ | $0.037_{\pm 0.000}$ | $1.361_{\pm 0.045}$ |
| + KD + ODS | | $94.13_{\pm 0.08}$ | $\mathbf{0.200}_{\pm 0.005}$ | $\mathbf{0.090}_{\pm 0.001}$ | $\mathbf{0.028}_{\pm 0.001}$ | $\mathbf{2.105}_{\pm 0.158}$ |
| + KD + ConfODS | | $\mathbf{94.18}_{\pm 0.12}$ | $0.205_{\pm 0.004}$ | $0.091_{\pm 0.001}$ | $0.029_{\pm 0.001}$ | $1.950_{\pm 0.066}$ |
| **BatchEns « DeepEns (WideResNet-28x10 on CIFAR-100)** | | | | | | |
| $\mathcal{T}$ : DeepEns-4 | 146.15 M | 82.52 | 0.676 | 0.250 | 0.035 | - |
| $\mathcal{S}$ : BatchEns-4 | 36.62 M | $80.34_{\pm 0.08}$ | $0.900_{\pm 0.005}$ | $0.298_{\pm 0.002}$ | $0.094_{\pm 0.001}$ | $0.972_{\pm 0.001}$ |
| + KD | | $80.51_{\pm 0.22}$ | $0.803_{\pm 0.010}$ | $0.283_{\pm 0.001}$ | $0.073_{\pm 0.005}$ | $1.000_{\pm 0.007}$ |
| + KD + Gaussian | | $80.39_{\pm 0.12}$ | $0.816_{\pm 0.010}$ | $0.286_{\pm 0.000}$ | $0.070_{\pm 0.003}$ | $0.994_{\pm 0.003}$ |
| + KD + ODS | | $\mathbf{81.88}_{\pm 0.32}$ | $0.680_{\pm 0.016}$ | $\mathbf{0.257}_{\pm 0.006}$ | $\mathbf{0.023}_{\pm 0.003}$ | $3.987_{\pm 1.325}$ |
| + KD + ConfODS | | $81.85_{\pm 0.32}$ | $\mathbf{0.678}_{\pm 0.009}$ | $\mathbf{0.257}_{\pm 0.003}$ | $\mathbf{0.023}_{\pm 0.001}$ | $\mathbf{3.990}_{\pm 0.770}$ |

**Table 9:** Cross-architecture knowledge distillation for a model compression on CIFAR-100 and TinyImageNet: ACC and *standard metrics* including NLL, BS, ECE, and DEE. Refer to Table 2 for the results with calibrated metrics.

| Method | # Params | ACC (↑) | NLL (↓) | BS (↓) | ECE (↓) |
|---|---|---|---|---|---|
| **BatchEns-4 « DeepEns-4 (WRN-28x2 on CIFAR-100)** | | | | | |
| $\mathcal{T}$ : DeepEns-4 | 146.15 M | 82.52 | 0.676 | 0.250 | 0.035 |
| $\mathcal{S}$ : BatchEns-4 | 1.50 M | $75.17_{\pm 0.27}$ | $1.245_{\pm 0.024}$ | $0.383_{\pm 0.004}$ | $0.141_{\pm 0.004}$ |
| + KD | | $75.19_{\pm 0.36}$ | $1.207_{\pm 0.021}$ | $0.377_{\pm 0.007}$ | $0.136_{\pm 0.005}$ |
| + KD + Gaussian | | $74.50_{\pm 0.17}$ | $1.247_{\pm 0.012}$ | $0.389_{\pm 0.004}$ | $0.137_{\pm 0.003}$ |
| + KD + ODS | | $\mathbf{76.03}_{\pm 0.22}$ | $\mathbf{0.899}_{\pm 0.013}$ | $\mathbf{0.333}_{\pm 0.003}$ | $\mathbf{0.027}_{\pm 0.002}$ |
| + KD + ConfODS | | $76.01_{\pm 0.16}$ | $0.901_{\pm 0.006}$ | $0.334_{\pm 0.002}$ | $0.028_{\pm 0.004}$ |
| **BatchEns-4 « DeepEns-4 (WRN-28x5 on CIFAR-100)** | | | | | |
| $\mathcal{T}$ : DeepEns-4 | 146.15 M | 82.52 | 0.676 | 0.250 | 0.035 |
| $\mathcal{S}$ : BatchEns-4 | 9.20 M | $78.75_{\pm 0.11}$ | $1.031_{\pm 0.022}$ | $0.324_{\pm 0.004}$ | $0.112_{\pm 0.003}$ |
| + KD | | $78.89_{\pm 0.10}$ | $0.932_{\pm 0.021}$ | $0.314_{\pm 0.004}$ | $0.095_{\pm 0.001}$ |
| + KD + Gaussian | | $78.80_{\pm 0.41}$ | $0.929_{\pm 0.018}$ | $0.313_{\pm 0.007}$ | $0.090_{\pm 0.006}$ |
| + KD + ODS | | $80.24_{\pm 0.05}$ | $0.744_{\pm 0.007}$ | $0.279_{\pm 0.002}$ | $0.026_{\pm 0.003}$ |
| + KD + ConfODS | | $\mathbf{80.62}_{\pm 0.25}$ | $\mathbf{0.735}_{\pm 0.007}$ | $\mathbf{0.275}_{\pm 0.003}$ | $\mathbf{0.023}_{\pm 0.001}$ |
| **BatchEns-4 « DeepEns-4 (WRN-28x5 on TinyImageNet)** | | | | | |
| $\mathcal{T}$ : DeepEns-4 | 146.40 M | 69.90 | 1.243 | 0.404 | 0.027 |
| $\mathcal{S}$ : BatchEns-4 | 9.23 M | 64.86 | 2.019 | 0.520 | 0.179 |
| + KD | | 65.86 | 1.782 | 0.493 | 0.149 |
| + KD + Gaussian | | 65.72 | 1.792 | 0.494 | 0.148 |
| + KD + ODS | | **65.98** | **1.440** | **0.460** | **0.059** |