# OpenReview forum: "Diversity Matters When Learning From Ensembles"
_NeurIPS.cc/2021/Conference — NeurIPS 2021 Poster_

### Official Review · Reviewer_z2qY · 2021-07-08

**Rating:** 7
**Confidence:** 3

**Summary:**

This paper focuses on the topic of effectively distilling ensembles of models into a single model. Assuming that the distilled model should inherit the diversity of the ensemble, they attempt to make the distilled performance more comparable to the ensemble by using a (adversarial) perturbation strategy that directly identifies sources of diversity within the ensemble (i.e., where models within the ensemble disagree with each other). The authors demonstrate empirically that utilizing such a diversity-inducing objective within distillation improves the final performance of the distilled model in comparison to the ensemble on image classification experiments. They motivate this objective for distillation with intuitive descriptions of the importance of diversity within an ensemble, as well as by making connections to gradient matching between the student and teacher.

**Limitations And Societal Impact:**

I address limitations within my main review (authors also address limitations in the paper directly). There is no negative societal impact.

**Main Review:**

I begin my review by emphasizing that I am completely open to authors' comments and feedback in regard to my review. My final score will be mostly based upon discussion with authors in regard to my questions and initial thoughts on the manuscript.

Pros:
- A reasonable explanation is provided for the gap in distilled network performance in comparison to the ensemble that is supported by evidence. I found this discussion to be intuitive and useful.
- This method reduces the significant overhead (memory and computation) of ensembles by only using a single model, but narrows the gap in performance between the model and the ensemble.
- Authors make a connection to gradient matching that is explained well and supported empirically. This is a nice interpretation/understanding of why the method works.
- The methodology seems to consistently improve the performance of distilled models in comparison to distillation with gaussian noise or normal distillation procedures.
- Paper is generally well-written, and it seems like the authors have spent time to polish it.

Cons:
- There is not much performance improvement on CIFAR10. It would be good to replace this with a larger dataset (or simply add a larger dataset).
- Only one method of input perturbation is attempted. It would be nice to have justification for why ODS is the best perturbation strategy and comparisons to other options in the experiments. I see that you show Gaussian perturbation does not induce diversity, but it seems that there should be numerous other perturbation strategies that would induce a similar result to ODS.
- No large-scale classification experiments (e.g., ImageNet). I believe running experiments on ImageNet for this work is important, as it is unclear from experiments whether your approach will scale to datasets with a significantly increased number of classes (e.g., 1000 instead of 100). This is especially true because a significant performance difference is seen between 10 and 100 class cases. It seems like ODS has been tested on ImageNet by Tashiro et. al., however.
- The experiments focus on one knowledge distillation strategy (one-to-one batch ensemble distillation). Is it reasonable to test some other methods of distillation with a similar ODS approach?


Questions:
- Are adversarial perturbations the most appropriate kind to use within this methodology? I am not fully convinced that adversarial perturbations will capture diversity in the ensemble that has tangible, semantic meaning, but rather small/imperceptible changes that drastically modify network output.
- Assumption that diversifying directions transfer between teachers is fully justified. (But, how do we choose which teacher to use? Does it matter?)

Small Comments:
- You probably need a section on adversarial attacks on neural networks in the related work, as the perturbation strategy for this work is inherited from an adversarial attack paper.
- It would be nice to provide some clearer explanation of why you choose the exact distillation strategy that you use. For me, it is not immediately clear whether using batch ensemble and one-to-one distillation is a reasonable choice (i.e., is this widely considered the SOTA approach?)


**Time Spent Reviewing:**

1.5

---

> ### Author Response · Authors · 2021-08-10
> **Response to Reviewer z2qY**
>
> > There is not much performance improvement on CIFAR10. It would be good to replace this with a larger dataset (or simply add a larger dataset).
>
> > No large-scale classification experiments (e.g., ImageNet). I believe running experiments on ImageNet for this work is important, as it is unclear from experiments whether your approach will scale to datasets with a significantly increased number of classes (e.g., 1000 instead of 100). This is especially true because a significant performance difference is seen between 10 and 100 class cases. It seems like ODS has been tested on ImageNet by Tashiro et. al., however.
>
> Please refer to our overall comment regarding our additional TinyImageNet experiments.
>
> * * *
>
> > Only one method of input perturbation is attempted. It would be nice to have justification for why ODS is the best perturbation strategy and comparisons to other options in the experiments. I see that you show Gaussian perturbation does not induce diversity, but it seems that there should be numerous other perturbation strategies that would induce a similar result to ODS.
>
> To the best of our knowledge, other commonly used perturbation strategies are adversarial perturbations, which are not necessarily suited to our problem setup of distilling from an ensemble of models. We chose ODS because its goal of diversifying outputs is closely aligned with our setup. Other than ODS, we further tried non-targeted adversarial perturbations and found it to be worse than ours (please refer to our comment for Reviewer 1b8g). We think that learning useful perturbations directions for diversification in a data driven way is an exciting future work.
>
> * * *
>
> > The experiments focus on one knowledge distillation strategy (one-to-one batch ensemble distillation). Is it reasonable to test some other methods of distillation with a similar ODS approach?
>
> We applied our method to MIMO (please refer to our comment for Reviewer 1b8g) and obtained significant improvement. We will put this result in the final version.
>
> * * *
>
> > Are adversarial perturbations the most appropriate kind to use within this methodology? I am not fully convinced that adversarial perturbations will capture diversity in the ensemble that has tangible, semantic meaning, but rather small/imperceptible changes that drastically modify network output.
>
> We chose ODS because of its very nature of increasing the diversity of the function outputs. We think that ODS is quite different from typical adversarial perturbations whose goal is to explicitly lower the prediction accuracy. As can be seen from the experiments using non-targeted adversarial perturbation instead of ODS (which actually harms the predictive performance), the usual adversarial perturbations are not well-suited for our purpose, and we conjecture that perturbed samples should have both diversities in outputs and low training losses in order to be beneficial for the distillation. So far, ODS (or ConfODS) is the only perturbation that we found to meet such criteria.
>
> * * *
>
> > Assumption that diversifying directions transfer between teachers is fully justified. (But, how do we choose which teacher to use? Does it matter?)
>
> Since they are on par with one another, we randomly choose one at each iteration.
>
> * * *
>
> > You probably need a section on adversarial attacks on neural networks in the related work, as the perturbation strategy for this work is inherited from an adversarial attack paper.
>
> Thank you for the suggestion. The next version will include a discussion of adversarial attacks in the related works section, and we will also place our method in the context of existing adversarial attack papers.
>
> * * *
>
> > It would be nice to provide some clearer explanation of why you choose the exact distillation strategy that you use. For me, it is not immediately clear whether using batch ensemble and one-to-one distillation is a reasonable choice (i.e., is this widely considered the SOTA approach?)
>
> We refer to two recent papers [Mariet et al., 2020] and [Tran et al., 2020] studying one-to-one distillation as a strong alternative for the vanilla ensemble. Another ensemble distillation approach would be EnD^2 for which we found to be less effective than ours.
>
> * * *
>
> References
> * [Mariet et al., 2020] Z. Mariet et al., “Distilling Ensembles Improves Uncertainty Estimates”, AABI 2021.
> * [Tran et al., 2020] L. Tran et et al., “Hydra: Preserving Ensemble Diversity for Model Distillation”, ICML 2020 Workshop on Uncertainty and Robustness in Deep Learning.

---

> > ### Comment · Reviewer_z2qY · 2021-08-20
> > **Response to rebuttal**
> >
> > Based on the authors response to my comments, I believe my main concerns: i) lack of experiments, ii) only using one perturbation method, and iii) choice of distillation strategy have been addressed. I would ask that the authors include their supplemental experiments, as well as provide their discussion of why ODS is the correct methodology to use and why they choose their exact distillation strategy within the paper. It seems that both points ii) and iii) above can be easily addressed if the explanation within the rebuttal is included in the paper. My only remaining concern is the lack of large-scale experiments, but TinyImageNet does provide larger-scale results that make the paper more convincing. I raised my score one point accordingly.

---

> > > ### Author Response · Authors · 2021-08-20
> > > **thanks**
> > >
> > > Thanks for your effort in reviewing our paper and re-evaluating it. As you suggested, we will move the additional experimental results and discussions on the choice of ODS as a perturbation strategy in the main text.

---

### Official Review · Reviewer_p6SF · 2021-07-13

**Rating:** 5
**Confidence:** 5

**Summary:**

This paper proposes to improve Knowledge Distillation (KD) for ensemble models. The authors apply ODS [Tashiro et.al, 2020] to each teacher model in the ensemble to obtain perturbed data. Thereafter, the perturbed data is used to train student models, each corresponding to one teacher model. The authors argue that ODS can improve the diversity among student models in the ensemble.

**Limitations And Societal Impact:**

Yes

**Main Review:**

The main contribution is applying an existing technique, ODS [Tashiro et.al, 2020] to a non-trivial problem of ensemble model distillation. The abstract (line 12) promises to find input samples that teacher models' outputs disagree. Then I would expect one has to feed all teacher models' outputs into the process of seeking the perturbed inputs. But algorithm 1 treats each teacher model separately in the process. It seems to me that the algorithm 1 completely relies on the diversifying ability of ODS, and assumptions made in line 113-114.

However, assumption in line 113-114 is quite counter-intuitive to me. Usually, a wise design of ensemble model is to include diversified member models. In this case, this means that the teacher models are very different from each other. If so, intuitively, a perturbation that maximumly change one teacher's decision does not necessarily change another's. For the similar reason, the assumption made on Jacobians (line 143) seems too strong, and most likely cannot hold in practice. This concern challenges the correctness of using one teacher model to derive perturbation vectors for all teacher models (line 110-111).

The paper has many clarity issues. Below is an incomplete list of them:

line 96-97: It is not very clear how ODS works. Which one is the perturbation vector? $w$ or $v_{ODS}$? I assume it is $v_{ODS}$, after checking the seminal work [Tashiro et.al, 2020]. Please improve the presentation, e.g., including equation (3) from the seminal paper.

Before Equation (9), please define $\epsilon$ as the perturbation in this specific context. Using multiple perturbation vectors, $v_{ODS}$ and $\epsilon$, may be hard for readers to understand.

Equation (10) is taken from equation (2) of [Srinivas and Fleuret, 2018]. It is therefore not this paper's contribution, and the authors should say that in the paper. Also, the notation is very informal. What does $\frac{J_s(x)}{S(x)}$ mean? a matrix divide by a vector?

Line 194-195: I cannot find a definition of "confidence value". Is it the predicted probability of the correct class label? Please define it first. Fig 3 seems to be very important, as it fulfills the promise made in abstract (line 12). However, the setup from line 192-198 is hard to read, which made it hard to judge the correctness and significance of fig.3.

Line 224: It is not clear what the ROC curve means. Also how is it obtained? I cannot find any description in the main paper or supplementary material.

Line 234-235, why higher SNR is more effective?

Overall, the paper does not make it crystal clear why diversity among teacher models are signified/promoted.

Through multiple rounds of Q&A, I think correctness is now justified, and I appreciate the authors' efforts in clarification. Nevertheless, the current draft needs to include all the key points during the discussion process. For example, the draft itself needs heavy revision to manifest how diversity among ensemble members is achieved. I now raise the score to 5, due to the fact that such a revision may require another round of reviewing.

**Time Spent Reviewing:**

10

---

> ### Author Response · Authors · 2021-08-10
> **Response to Reviewer p6SF**
>
> > However, assumption in line 113-114 is quite counter-intuitive to me. Usually, a wise design of ensemble model is to include diversified member models. In this case, this means that the teacher models are very different from each other. If so, intuitively, a perturbation that maximumly change one teacher's decision does not necessarily change another's. For the similar reason, the assumption made on Jacobians (line 143) seems too strong, and most likely cannot hold in practice. This concern challenges the correctness of using one teacher model to derive perturbation vectors for all teacher models (line 110-111).
>
> We respectfully disagree with this comment, both intuitively and empirically. While it is true that the diversity of ensemble models is a crucial factor for their robustness, the ensemble models are not “very” different from each other. After all, they are trained on the same dataset using the same model and hyperparameters. After achieving zero training error, most of their predictions for the training set agree with each other. The diversity arises when applied to the test set, yet they mostly agree except for minor class probabilities. This disagreement in minor class probabilities actually brings robustness to the prediction. In this sense, it is not counter-intuitive to assume that the perturbation obtained from ODS or the gradients evaluated from a single model might transfer to the other model.
>
> Furthermore, our assumption in lines 113-114 is verified by empirical results in prior work and our own paper. Experiments in the original ODS paper [Tashiro et al., 2020] showed that diversified examples generated with ODS can be transferred from a source model to a target model (Appendix D.1). Even aside from ODS, this assumption is commonly made in the black-box adversarial attack literature. Please also note that our experiments in Sections 5.2-5.3 explicitly reconfirm this assumption in the context of distilling DE. More specifically, we measure mean KLD values (Fig 6a) and cosine similarity (Fig 6b), revealing that diversifying directions transfer between teacher models.
>
> Our original submission referenced experiments in the original ODS paper in lines 114-115 and our own experiments in line 116. We note that reviewer z2qY found our evidence sufficient: “Assumption that diversifying directions transfer between teachers is fully justified.” In the next version, we will more thoroughly discuss the relation of our assumption to the black-box attack literature, in addition to the meaning of our empirical results.
>
> * * *
>
> > Line 224: It is not clear what the ROC curve means. Also how is it obtained? I cannot find any description in the main paper or supplementary material.
>
> We apologize for the confusion. The ROC curves compare (1) the distribution of cosine similarities of gradients on training (or validation) examples between DE teacher and corresponding BE student, and (2) the distribution of cosine similarities of gradients on training (or validation) examples between DE teacher and the baseline BE model trained without knowledge distillation (i.e., $\alpha=0$). Hence, higher AUROC indicates better separation between the gradients of student models with/without perturbations during distillation, and the results shows that KD + ODS clearly improves the gradient matching compared to the baselines. We will include a self-contained description of the ROC curves in the next version.
>
> * * *
>
> > Line 234-235, why higher SNR is more effective?
>
> Let (12)=$\text{GN}(\mathbf{x},\mathbf{\epsilon})$. Then, we defined $\text{SNR}(\mathbf{x})=\frac{\lVert{\mathbb{E}\_{\epsilon} \left[\text{vec} (\text{GN}(\mathbf{x},\epsilon))\right]}\rVert\_2}{\sqrt{\lVert{\text{Var}\_{\epsilon} \left[\text{vec}(\text{GN}(\mathbf{x},\epsilon))\right]}\rVert\_2}}$ where $\text{vec}(\mathbf{x})$ is the vectorization of $\mathbf{x}$ and $\text{Var}[X]$ is the element-wise variance of $X$. A higher SNR denotes that the gradient is more consistent (less stochastic) with perturbations, making the learning dynamics more stable and efficient. Note that the SNR is a standard metric for assessing the quality of stochastic gradients, e.g., [Rainforth et al., 2018] and [Tucker et al., 2019].
>
> * * *
>
> > line 96-97: It is not very clear how ODS works. Which one is the perturbation vector? $w$ or $v_{ODS}$? I assume it is $v_{ODS}$, after checking the seminal work [Tashiro et.al, 2020]. Please improve the presentation, e.g., including equation (3) from the seminal paper.
>
> We apologize for the confusion. Equation (6) in Section 3.1 describes that $\mathbf{v}_{\text{ODS}}$ is the perturbation vector which is used to generate perturbed input $\mathbf{\tilde{x}}$, and $\mathbf{w}$ is the randomized weight imposed to each class of $\mathcal{F}(\mathbf{x})$ that induces diversification of perturbation vector.
>
> * * *
>
> > Before Equation (9), please define $\epsilon$ as the perturbation in this specific context. Using multiple perturbation vectors, $v_{ODS}$ and $\epsilon$, may be hard for readers to understand.
>
> We will revise the document as you suggested.
>
> * * *
>
> > Equation (10) is taken from equation (2) of [Srinivas and Fleuret, 2018]. It is therefore not this paper's contribution, and the authors should say that in the paper.
>
> Please note that we already gave the credit by citing Srinivas and Fleuret [2018] in lines 126-132. We will further clarify this in the next version.
>
> * * *
>
> > Also, the notation is very informal. What does $\frac{J_{S}(x)}{S(x)}$ mean? a matrix divide by a vector?
>
> We apologize for the abuse of notation. It should be written as $
> \mathcal{L}\_{\text{JM}}\left( \mathcal{S}(x),\mathcal{T}(x) \right) = -\mathbb{E}\_{\varepsilon}\left[ \sum\_{k=1}^{K} \frac{ \varepsilon^\top \nabla\_{x}\mathcal{T}^{(k)}(x) \nabla\_{x}^{\top}\mathcal{S}^{(k)}(x) \varepsilon }{ \mathcal{S}^{(k)}(x) } \right]$. In other words, the sum of ratios between $k$th components. We will make this clearer in the next version.
>
> * * *
>
> > Line 194-195: I cannot find a definition of "confidence value". Is it the predicted probability of the correct class label? Please define it first. Fig 3 seems to be very important, as it fulfills the promise made in abstract (line 12). However, the setup from line 192-198 is hard to read, which made it hard to judge the correctness and significance of fig.3.
>
> We apologize for the confusion.  The “confidence” of a model is the class probability of the predicted label, meaning the maximum class probability. The diversity plot shows the minimum confidence across all teacher models, binned according to value. We will further clarify this in the next version.
>
> * * *
>
> References
> * [Rainforth et al., 2018] Tighter variational bounds are not necessarily better, ICML 2018
> * [Tucker et al., 2019] Doubly reparameterized gradient estimators for Monte Carlo objectives, ICLR 2019.
> * [Tashiro et al., 2020] Y. Tashiro et al., “Diversity Can Be Transferred: Output Diversification for White- and Black-box Attacks”, NeurIPS 2020

---

> > ### Comment · Reviewer_p6SF · 2021-08-20
> > **Details still missing, correctness still unjustified**
> >
> > I appreciate the authors' efforts in clarification. However, the following questions remain unanswered.
> >
> > 1) I understand that the authors want to show that with the ODS perturbation, gradients of student better matches those of the teacher. However, it is still not clear to me how a ROC curve (in fig. 5) is obtained. When plotting a ROC, usually we want to distinguish between two distributions, $p_1(a)$ and $p_2(a)$. In this case, what are $p_1(a)$ and $p_2(a)$? What is the $a$?
> >
> > 2) A new question: in Eq (11), do you mean $NLL(\mbox{Ensemble of }\ell \mbox{ models}) \leq NLL(\mathcal S)$? Correct me if I'm wrong. In this case, you hold the (learned) student model $\mathcal S$ fixed, and use some another ensemble of $\ell$ models to match the performance of $\mathcal S$. Then, how are these $\ell$ models obtained?
> >
> > 3) Again, my biggest concern is, I cannot see why the proposed method could improve disagreement/diversity among ensemble members, although fig.3 tries to validate that. In particular, for an input image $\mathbf x$, ODS promotes the diversity among $f(\mathbf x+\mathbf v_{ODS}^n), n=1,\dots, N$. Here $f(\cdot)$ denotes output decision of a network, and $\mathbf v_{ODS}^n$ denotes multiple ODS perturbation vectors. In addition, the transferability (in the seminal ODS paper) means one could use another well-trained network $g(\cdot)$ to compute the $v_{ODS}^n, n=1,\dots$, and still obtained diversified $f(\mathbf x+\mathbf v_{ODS}^n)$. On the other hand, in this paper, we really need the disagreement among ensemble members. The notion of diversity is different! I don't see this justified in the authors' response.

---

> > > ### Author Response · Authors · 2021-08-21
> > > **Response to Reviewer p6SF (#2)**
> > >
> > > We thank you for your time to review our article.
> > >
> > > > I understand that the authors want to show that with the ODS perturbation, gradients of student better matches those of the teacher. However, it is still not clear to me how a ROC curve (in fig. 5) is obtained. When plotting a ROC, usually we want to distinguish between two distributions, $p_1(a)$ and $p_2(a)$. In this case, what are $p_1(a)$ and $p_2(a)$? What is the $a$?
> > >
> > > $p_1(a)$ would be the distribution of the cosine similarities between the gradients of a teacher network and a student network when the student network is trained by knowledge distillation (+ Gaussian or ODS perturbation). $p_2(a)$ is then the distribution of the cosine similarities between gradients of a teacher network and a student network when the student network is trained from scratch (without any guide from the teacher network). As shown in Figure 5, vanilla KD and KD + Gaussian promotes gradient matching to some extent, meaning that $p_1(a)$ is shifted to the right compared to $p_2(a)$, so separating those two distributions. However, even if KD + Gaussian explicitly encourages gradient matching (equation (10)), the actual gain from it is not significant compared to vanilla KD. On the other hand, KD + ODS significantly improves the separation (higher AUROC value). Therefore, we think drawing the empirical distributions of $p_1(a)$ and $p_2(a)$ would be a better way to present the result, so we will revise this in the final version of the paper.
> > >
> > > > A new question: in Eq (11), do you mean $NLL(\text{Ensemble of } \ell \text{ models}) \leq NLL(\mathcal{S})$? Correct me if I'm wrong. In this case, you hold the (learned) student model $\mathcal{S}$ fixed, and use some another ensemble of $\ell$ models to match the performance of $\mathcal{S}$. Then, how are these $\ell$ models obtained?
> > >
> > > We apologize for the typo. As you pointed out, the inequality should be $\leq$. The ensemble models are obtained by actually training them. We already have those models when we are constructing teacher networks. More specifically, given a maximum $M$ of trained teacher networks, we compute NLL values for $\ell=1, 2, \dots, M$ models and use them to evaluate DEE values (NLL values for non-integer $\ell$ are obtained by linear interpolation).
> > >
> > > > Again, my biggest concern is, I cannot see why the proposed method could improve disagreement/diversity among ensemble members, although fig.3 tries to validate that. In particular, for an input image $\mathbf{x}$, ODS promotes the diversity among $f(\mathbf{x} + \mathbf{v}\_{ODS}^{n}), n=1,...,N$. Here $f(\cdot)$ denotes output decision of a network, and $\mathbf{v}\_{ODS}^{n}$ denotes multiple ODS perturbation vectors. In addition, the transferability (in the seminal ODS paper) means one could use another well-trained network $g(\cdot)$ to compute the $v\_{ODS}^{n}, n=1,...$, and still obtained diversified $f(\mathbf{x} + \mathbf{v}\_{ODS}^{n})$. On the other hand, in this paper, we really need the disagreement among ensemble members. The notion of diversity is different! I don't see this justified in the authors' response.
> > >
> > > Thank you for raising this question; we initially thought that the transferability property demonstrated in the original ODS paper naturally leads to this conclusion, but we now realize that our reasoning may not have been straightforward to readers. We will explain our intuition and also supplement it with additional arguments which we believe clarify this issue.
> > >
> > > Please refer to Figure 1 of Tashiro et al., 2020. The third and fourth boxes show how ODS perturbations computed from a surrogate model act on a target model. Due to the difference in the gradient between the surrogate and target model, the same ODS perturbation moves outputs of the surrogate model and the target model to different directions. This disagreement is what we are looking for; when an ODS perturbation is computed from a specific teacher (surrogate), it will behave differently to another teacher (target).
> > >
> > > More specifically, let $f_1(x)$ and $f_2(x)$ be two teacher networks, and $v_\text{ODS} = w^\top \nabla f_1(x)$ be an ODS perturbation computed from the first teacher (omitting the normalization constant for simplicity). Let $\nabla f_1(x) \approx R \nabla f_2(x)$ for some matrix $R$. Then we have $v_\text{ODS} = w^\top (R \nabla f_2(x)) = (R^\top w)^\top \nabla f_2(x)$. That is, the same $v_\text{ODS}$ tries to move $f_1(x)$ towards $w$ and $f_2(x)$ towards $R^\top w$ (as you have written down, using different ODS samples (with $w$ and $R^\top w$) leads to the diversities in the predictions). Due to the transferability assumption, we can assume that $\nabla f_1(x)$ and $\nabla f_2(x)$ are quite similar ($R$ is close to identity), meaning that $w$ and $R^\top w$ to be also similar, but still different enough to simulate the actual diversity of the deep ensemble teachers. Note that without the transferability assumption, the directions of $w$ and $R^\top w$ can be vastly different, meaning that the resulting predictions of $f_1(x)$ and $f_2(x)$ do not agree at all. As we have empirically shown (see our comments for reviewer z2qY), the examples with high training loss (i.e. the ones with low uncertainties or the ones for which most of the ensemble teachers have different predictions) are not helpful for distillation even if they have high diversities.
> > >
> > > Another argument we found to be useful is as follows. Ideally, we want to find a direction that can cause disagreement between the teacher networks, so we want to compute $v_\text{ODS-all} = \sum_{j=1}^M w_j \nabla f_j(x)$ where $w_1, \dots, w_M \sim \mathrm{Unif}([-1, 1])$. Computing this would be costly because we need to evaluate $M$ gradients of the teacher networks. Our approach of choosing a random teacher and computing $v_\text{ODS} = w_j \nabla f_j(x)$ for a uniformly drawn $j$, can be understood as a stochastic gradient approximation of the full gradient $v_\text{ODS-all}$. During our algorithm development phase, we checked whether this approximation is effective, and we found that the performance from using $v_\text{ODS}$ is almost identical to using $v_\text{ODS-all}$. For instance, the following result is an early-stage experiment on CIFAR-10, which we didn’t include in the paper (note that this is not directly comparable to results in the paper since they use different hyperparameters).
> > >
> > > | Method | ACC | NLL | NLL (calibrated) |
> > > | :- | :- | :- | :- |
> > > | DE-4 Teacher                              | 0.9473 | 0.1619 | 0.1606
> > > | BE-4 Student                              | 0.9349 | 0.2757 | 0.1986
> > > | + KD                                            | 0.9417 | 0.2340 | 0.1797
> > > | + KD + Gaussian                          | 0.9411 | 0.2344 | 0.1808
> > > | + KD + ODS (use one random T) | 0.9422 | 0.1974 | 0.1732
> > > | + KD + ODS-all (use all Ts)          | 0.9415 | 0.1961 | 0.1753
> > >
> > > As you can see from the result, the performance of KD + ODS (with $v_\text{ODS}$) is almost identical to that of KD + ODS-all (with $v_\text{ODS-all}$). We conjecture that this is due to the transferability of the gradients: because the gradients $\{\nabla f_j(x)\}_{j=1}^M$ are quite similar to each other, the variance of the stochastic gradient $v_\text{ODS}$ is quite low, making it an effective estimate. Without the transferability, the $v_\text{ODS}$ would be a noisy gradient estimate leading to poor performance.
> > >
> > > We hope these arguments clarify your concerns. We will include them in the revised version of the paper.

---

> > > > ### Comment · Reviewer_p6SF · 2021-08-21
> > > > **Concerns mostly resolved**
> > > >
> > > > The authors' explanation has mostly addressed my concerns, although I think the paper at current stage needs quite some revision, e.g., clarify the notion of diversity (among teacher networks), and where the diversity comes from. Overall, I think the authors should put the reasonings in response into the paper, before introducing ODS.
> > > >
> > > > I actually find the description of ODS confuses readers, e.g., line 100, which seems to suggest diversity among ensemble members comes from multiple random $\mathbf w$'s, but in fact it is not. It is really because of the difference between $\nabla f_1(\mathbf x)$ and  $\nabla f_2(\mathbf x)$ (as clarified in the response). Due to this reason, I also think that it is not necessary to talk about transferability (of diversity). There is no transfer here.
> > > >
> > > > The rebuttal makes me re-evaluate this paper as a borderline paper. I'm glad to raise the score to 5, as a major revision will require another round of review.

---

> > > > > ### Author Response · Authors · 2021-08-21
> > > > > **Response to Reviewer p6SF (#3)**
> > > > >
> > > > > We thank you for re-evaluating our work and suggestion to enhance the presentation. We fully agree with you that we should first clearly explain what diversity we are looking for and how ODS can achieve this. We will put the discussion in the response to the paper, and rephrase the text to reduce confusion.
> > > > > As far as we know, we cannot submit a revision during the discussion period. It would be greatly appreciated if you could raise the score in the current system, and we will do our best to edit the paper as you suggested if our paper is accepted.

---

### Official Review · Reviewer_7xWm · 2021-07-16

**Rating:** 7
**Confidence:** 4

**Summary:**

This paper attempts to reduce the performance gap between an ensemble and a model that was distilled from it. The main idea is that, when distilling, the model should “absorb as much function diversity inside the ensemble as possible”. To do that, the paper proposes using ODS perturbation strategy, that samples diverse predictions for a given input form the ensemble, to use as label during distillation.


**Limitations And Societal Impact:**

Yes

**Main Review:**

The ideas proposed are neat and well motivated by previous literature finding that prediction diversity is important for ensemble performance. I found particularly interesting the results that ODS sampling is approximately gradient matching, and that by doing gradient matching, the distilled model is encouraged to behave like the teachers on “nearby points”.

My main concern with the paper is that most of the experiments are performed on CIFAR-10 and CIFAR-100. I would like to see experiments on a more realistic benchmark, such as ImageNet, to assess the generality of the conclusions.

I would also be interested in seeing other ensembling methods tested, such as EMA.

Otherwise, I found the paper to be easy to read, well explained, and the analyses to be insightful.


**Time Spent Reviewing:**

2

---

> ### Author Response · Authors · 2021-08-10
> **Response to Reviewer 7xWm**
>
> > My main concern with the paper is that most of the experiments are performed on CIFAR-10 and CIFAR-100. I would like to see experiments on a more realistic benchmark, such as ImageNet, to assess the generality of the conclusions.
>
> Please refer to our overall comment regarding our additional TinyImageNet and corrupted dataset experiments.
>
> * * *
>
> > I would also be interested in seeing other ensembling methods tested, such as EMA.
>
> We chose DE as our base ensemble method because of its simplicity and impressive empirical results reported in prior works. Even though we used DE for our experiments, the choice of ensemble method is orthogonal to our method, which distills the ensemble into a single network.
>
> As you suggested, we may also try EMA as an alternative ensembling method. Other alternatives are fast geometric ensembles [Garipov et al., 2018] and stochastic weight averaging [Izmailov et al., 2018]. We think investigating which ensemble method is best for downstream distillation is an exciting research direction.
>
> * * *
>
> References
> * [Garipov et al., 2018] T. Garipov et al., “Loss Surfaces, Mode Connectivity, and Fast Ensembling of DNNs”, NeurIPS 2018.
> * [Izmailov et al., 2018] P. Izmailov et al., “Averaging Weights Leads to Wider Optima and Better Generalization”, UAI 2018.

---

> > ### Comment · Reviewer_7xWm · 2021-08-20
> > **re: rebuttal**
> >
> > Post-rebuttal comments: I thank the authors for their time on the rebuttal and on the additional experiments. I maintain my original score.

---

> > > ### Author Response · Authors · 2021-08-20
> > > **thanks**
> > >
> > > Thanks for your effort to review our paper and encouraging comment.

---

### Official Review · Reviewer_1b8g · 2021-07-18

**Rating:** 7
**Confidence:** 4

**Summary:**

The paper studies the question of ensemble function diversity and how to preserve it on ensemble distillation to a single model. The authors use input perturbations that make the models, especially well trained models that are near 0 loss on the training set, disagree on the input, and then use knowledge distillation from individual members of the ensemble to individual subnetworks of the distilled model. The authors show that without this approach, the distilled models suffer from lack of function diversity, and that their approach enhances it and in accordance with their theory enhances performance as well.

**Limitations And Societal Impact:**

Yes

**Main Review:**

I like the paper. It is well written, deals with an important question that is of interest to the community, and presents its approach in an easy to understand way. A small caveat is that despite following the subfield pretty closely I might not be able to judge the originality of the approach to the full extent and will likely defer to other reviewers if they have substantial points on this.

STRONG ASPECTS:
1) the paper is very clearly presented and was easy to read and understand
2) the problem the authors are looking at and addressing is important and broadly practically relevant
3) the technique is simple and could be implemented relatively easily (maybe apart from the gradient-informed input perturbation, which requires access to the models in the ensemble to be distilled)
4) I like the gradient matching perspective on why this technique could be working well

QUESTIONS AND CLARIFICATIONS:
1) what other approach to making sure that function diversity is transferred to the distilled model are there? did you compare your technique to them?
2) In Figure 2 and in the subsequent text it seems to be assumed that you are distilling an ensemble of models to a single model that still has a distinct number of subnetworks but is cheaper (for example the batch ensemble you are discussing). Could you approach work for actually distilling to a single model without this explicit multiple-subnetworks structure.
3) What is the advantage of using the gradient-informed perturbations in Equation 5 over the uniform perturbations? You mention that the random perturbations will be bad if the underlying data is low dimension,which I understand. This paper https://openreview.net/forum?id=XMoyS8zm6GA seems to suggest that they are actually very high dimension, which might make the random perturbations similarly good?
4) Could you perhaps use a MIMO-like architecture to be distilled into as well? I found this one by Havasi et al to be pretty high performance https://arxiv.org/abs/2010.06610 (only if you feel this might be a good idea, I know this is beyond the scope of your paper)
5) Aren't the perturbations in Eq. 5 pretty similar to adversarial perturbations, just with a target "label" being the random vector w?

WEAKER ASPECTS:
1) the experimental verification would be more convincing if you also looked at ImageNet. I know that this can be though, and I won't be decreasing my score based on it, but it could help a lot with the future adoption of your technique.
2) In Table 1 the models on CIFAR-100 reach only low 80% test accuracy. I think it would be good to show that your conclusions also work for more powerful SOTA models like the ViT that easily reach 90%+ test.

Overall, I like the paper!

UPDATE AFTER REBUTTAL:
I am very happy with the authors' detailed response and would like to increase my score from 6 to 7 to reflect that. Thank you very much and great job on the paper!



**Time Spent Reviewing:**

4

---

> ### Author Response · Authors · 2021-08-10
> **Response to Reviewer 1b8g**
>
> > what other approach to making sure that function diversity is transferred to the distilled model are there? did you compare your technique to them?
>
> This is an important point. To address it more thoroughly, we performed an additional experiment comparing to the recently proposed Ensemble Distribution Distillation (EnD^2) [Malinin et al., 2020], which considers the distribution of the predictions from deep ensemble (DE) teachers. As shown in the following table, ours significantly outperforms EnD^2 on ResNet32 with CIFAR10 setting in both accuracy and uncertainty estimation. All the uncertainty estimate metrics were measured at temperature scaled with the validation set.
>
> | Teacher | EnD^2 (ACC / NLL / BS / ECE) | Ours (ACC / NLL / BS / ECE)
> | :- | :- | :- |
> | DE-4 | 92.10% / 0.260 / 0.121 / 0.017 | 94.01% / 0.180 / 0.093 / __0.007__
> | DE-8 | 93.11% / 0.225 / 0.104 / __0.013__ | __94.18%__ / __0.174__ / __0.091__ / __0.007__
> | DE-16 | 93.21% / 0.221 / 0.104 / 0.016 | N/A
> | DE-32 | __93.49%__ / __0.218__ / __0.102__ / 0.014 | N/A
>
> We will include this baseline comparison in the next version.
>
> * * *
>
> > In Figure 2 and in the subsequent text it seems to be assumed that you are distilling an ensemble of models to a single model that still has a distinct number of subnetworks but is cheaper (for example the batch ensemble you are discussing). Could you approach work for actually distilling to a single model without this explicit multiple-subnetworks structure.
>
> > Could you perhaps use a MIMO-like architecture to be distilled into as well? I found this one by Havasi et al to be pretty high performance https://arxiv.org/abs/2010.06610 (only if you feel this might be a good idea, I know this is beyond the scope of your paper)
>
> Thank you for this suggestion; we agree that testing on architectures other than BE can reveal whether our strategy is generally applicable for distilling from ensembles. As you suggested, we performed additional experiments using the MIMO architecture [Havasi et al., 2020]. Even though MIMO constructs its multiple subnetworks implicitly (unlike BE), it can still be trained using our method. We applied KD from an ensemble of three teachers to MIMO-3 students with Gaussian and ODS perturbations using WRN28x10 and CIFAR100. The following table summarizes the results. All the uncertainty estimate metrics were measured at temperature scaled with the validation set. Again, ours significantly improved the performance compared to vanilla KD or KD with Gaussian perturbation.
>
> | Model | ACC / NLL / BS / ECE |
> | :- | :- |
> T : DeepEns-3 | 82.43% / 0.677 / 0.251 / 0.025 |
> S : MIMO-3 | 80.63% / 0.716 / 0.270 / __0.014__ |
> \+ KD | 80.75% / 0.720 / 0.271 / 0.021 |
> \+ KD + Gaussian | 80.69% / 0.723 / 0.271 / 0.016 |
> \+ KD + ODS | __81.17%__ / __0.692__ / __0.267__ / 0.020 |
>
> This result demonstrates that our proposed method is broadly beneficial independently of multi-network architecture and is not specific to BatchEnsemble. We will include this experiment in the next version.
>
> * * *
>
> > What is the advantage of using the gradient-informed perturbations in Equation 5 over the uniform perturbations? You mention that the random perturbations will be bad if the underlying data is low dimension,which I understand. This paper https://openreview.net/forum?id=XMoyS8zm6GA seems to suggest that they are actually very high dimension, which might make the random perturbations similarly good?
>
> We agree that the relationship between our method and the so-called “manifold hypothesis” is an important discussion point. First note that the manifold hypothesis is an informal statement with room for interpretation. As there is no standard mathematical definition of the data manifold, its dimensionality can differ based on measurement protocol.
>
> Reviewer 1b8g references [Fort et al., 2021], which studies the dimensionality of neural network class manifolds. This paper does suggest that the data manifold has a high dimension in general, but they also present experiments showing that model ensembling lowers this dimensionality (pg 7~8, Section 4.8), which is consistent with our findings. Indeed, our experiments (Section 5.3) show that gradient-informed perturbations have a much higher SNR value than uniform perturbation, suggesting that the manifold underlying DE has low dimensionality.
>
> Our paper can serve as a useful datapoint favoring low dimensionality for model ensembles in the ongoing dialogue about the manifold hypothesis for large datasets and deep neural networks. A simple intuition also discussed in our paper is that ODS’s gradient-informed perturbation is much more informative than random perturbation because our goal is to match the gradient, and the ODS perturbation contains information about one of these gradients. In the next version, we will discuss this issue in greater detail.
>
> * * *
>
> > Aren't the perturbations in Eq. 5 pretty similar to adversarial perturbations, just with a target "label" being the random vector w?
>
> As you pointed out, the ODS perturbation is very similar to the adversarial perturbation; if we replace the uniform vector $w$ with the one-hot class labels, we get an adversarial perturbation. The difference is that the adversarial perturbation is meant to worsen the predictive performance by design because it takes a step toward the directions increasing the classification loss. We empirically found that the perturbations just increasing diversity without maintaining prediction accuracy can actually harm the performance of student models (this is also related to the performance gain of ConfODS). Using ResNet32 on CIFAR10, we tested the KD with adversarial perturbations (i.e., non-targeted attack) instead of ODS perturbations. This achieved ACC of 93.32±0.22% and NLL of 0.194±0.003, which is much worse than vanilla KD with no perturbation.
>
> * * *
>
> > the experimental verification would be more convincing if you also looked at ImageNet. I know that this can be though, and I won't be decreasing my score based on it, but it could help a lot with the future adoption of your technique.
>
> > In Table 1 the models on CIFAR-100 reach only low 80% test accuracy. I think it would be good to show that your conclusions also work for more powerful SOTA models like the ViT that easily reach 90%+ test.
>
> Please refer to our overall comment regarding our additional TinyImageNet experiments.
>
> * * *
> References
> * [Malinin et al., 2020] A. Malinin et al., “Ensemble Distribution Distillation”, ICLR 2020.
> * [Havasi et al., 2020] M. Havasi et al., “Training independent subnetworks for robust prediction”, ICLR 2020.
> * [Fort et al., 2021] S. Fort et al., “Slice, Dice, and Optimize: Measuring the Dimension of Neural Network Class Manifolds”, 2021.

---

> > ### Comment · Reviewer_1b8g · 2021-08-31
> > **A response to a response**
> >
> > I am very happy with the authors' detailed response and would like to increase my score to reflect that. Thank you very much and great job on the paper!

---

> > > ### Author Response · Authors · 2021-09-01
> > > **Thanks**
> > >
> > > We thank you for your reevaluation and positive comments.

---

### Author Response · Authors · 2021-08-10
**Overall Response to All Reviewers**

We thank all reviewers for their thoughtful and constructive comments. Most reviewers were happy with the originality and significance of our work, while some concerns about the experiments were raised. We address some common concerns below and discuss more specific comments in individual responses addressed to each reviewer.

A common concern among the reviewers was the scalability of our method due to the lack of large-scale experiments. Although we could not complete ImageNet experiments within our time budget, we further tested the scalability of our method on the TinyImageNet dataset. This dataset consists of 100,000 images of size 64x64 from 200 classes, which we think is sufficiently representative of larger datasets. We provide a detailed description of this experiment and its results in section B.1 and Table 9 of the supplementary material, respectively. We believe our experiments demonstrate the scalability of our method, and we see no reason why it wouldn’t work on ImageNet and powerful SOTA models such as ViT. To make this more apparent, we will move the additional TinyImageNet experiments to the main text in the next version. Additionally, in section B.2 of the supplementary material, we also evaluate our method on corrupted datasets (CIFAR10-C and CIFAR100-C) to test our method on a more realistic data distribution. Compared to the baselines, our method is demonstrated to be much more robust to distributional shift.

---

### Decision · Program_Chairs · 2021-09-27

**Decision:**

Accept (Poster)

**Comment:**

The authors propose a method for distilling an ensemble to a single model by training on inputs which are perturbed to make ensemble member outputs disagree. Most reviewers (1b8g, 7xWm, z2qY) seemed to agree that the method was sensible, elegant, and clearly-presented, and that the gradient-matching intuition was clever and well-supported. Reviewer p6SF raised significant concerns about correctness of the authors' transferability assumption, but after rebuttal and discussion these were mostly addressed. p6SF also raised concerns about clarity, but these too were partially addressed and not in my opinion sufficient to justify rejection. Another common concern (1b8g, 7xWm, z2qY) was the lack of large-scale experiments, but the authors address this with TinyImageNet experiments in rebuttal to the satisfaction of multiple reviewers (1b8g, z2qY). Therefore I recommend acceptance.